# Epigenetic remodeling by vitamin C potentiates plasma cell differentiation

**Heng-Yi Chen**[1†], **Ana Almonte-Loya**[1,2†], **Fang-Yun Lay**[1], **Michael Hsu**[1], **Eric Johnson**[2], **Edahí González-Avalos**[2], **Jieyun Yin**[2], **Richard S Bruno**[3], **Qin Ma**[4,5], **Hazem E Ghoneim**[1,5], **Daniel J Wozniak**[1], **Fiona E Harrison**[6], **Chan-Wang Jerry Lio**[1,2,5]*

[1]Department of Microbial Infection and Immunity, The Ohio State University, Columbus, OH, United States; [2]Division of Gene Expression and Signaling, La Jolla Institute for Immunology, San Diego, CA, United States; [3]Human Nutrition Program, The Ohio State University, Columbus, OH, United States; [4]Biomedical Informatics, The Ohio State University, Columbus, OH, United States; [5]Pelotonia Institute for Immuno-Oncology, The James Comprehensive Cancer Center, College of Medicine, The Ohio State University, Columbus, OH, United States; [6]Department of Medicine, Vanderbilt University Medical Center, Nashville, TN, United States

**\*For correspondence:**
lio.4@osu.edu

[†]These authors contributed equally to this work

**Competing interest:** The authors declare that no competing interests exist.

**Abstract** Ascorbate (vitamin C) is an essential micronutrient in humans. The severe chronic deficiency of ascorbate, termed scurvy, has long been associated with increased susceptibility to infections. How ascorbate affects the immune system at the cellular and molecular levels remained unclear. From a micronutrient analysis, we identified ascorbate as a potent enhancer for antibody response by facilitating the IL-21/STAT3-dependent plasma cell differentiation in mouse and human B cells. The effect of ascorbate is unique as other antioxidants failed to promote plasma cell differentiation. Ascorbate is especially critical during early B cell activation by poising the cells to plasma cell lineage without affecting the proximal IL-21/STAT3 signaling and the overall transcriptome. As a cofactor for epigenetic enzymes, ascorbate facilitates TET2/3-mediated DNA modification and demethylation of multiple elements at the *Prdm1* locus. DNA demethylation augments STAT3 association at the *Prdm1* promoter and a downstream enhancer, thus ensuring efficient gene expression and plasma cell differentiation. The results suggest that an adequate level of ascorbate is required for antibody response and highlight how micronutrients may regulate the activity of epigenetic enzymes to regulate gene expression. Our findings imply that epigenetic enzymes can function as sensors to gauge the availability of metabolites and influence cell fate decisions.

## Editor's evaluation

This article describes the role of vitamin C in promoting plasma cell differentiation by remodeling the epigenome via TET family proteins. Activated Tet2/Tet3 actively demethylate selected genomic regions, including the genetic locus encoding Blimp-1, the master transcription factor of plasma cell differentiation. This article will be of interest to scientists in molecular immunologists, particularly those involved in epigenetic mechanisms of B cell differentiation to plasma cells.

## Introduction

Vitamin C (VC) or ascorbate is an essential micronutrient for maintaining cell barrier integrity and protecting cells from oxidative damage (*Maggini et al., 2007*; *Webb and Villamor, 2007*). Due to a mutation in *GULO* (L-gulono-gamma-lactone oxidase), humans are unable to synthesize ascorbate

and depend on dietary sources to achieve VC adequacy (**Nishikimi et al., 1994**). Long-term ascorbate deficiency leads to a disease termed *scurvy*, which has been associated with weakened immune responses and higher susceptibility to pneumonia (**Baron, 2009**). Although *scurvy* is rare nowadays, national surveillance data indicate that overt VC deficiency and suboptimal VC status occurs in ~7.1% and ~25% of adults in the US respectively (**Schleicher et al., 2009**). VC insufficiency is even more prevalent in some subgroups, including the elderly, smokers, those with limited dietary intake, and those with increased oxidative stress due to illnesses (**Isola et al., 2019**; **Cahill and El-Sohemy, 2010**; **Lima de Araújo et al., 2012**; **Na et al., 2006**). VC deficiency or insufficiency may contribute to the variation of immune responses against infections. Thus, it is critical to understand how VC regulates immune responses at the cellular and molecular level.

In past decades, VC has been examined as a therapeutic treatment for diseases, such as infections and cancers, but the results have been mixed (**Nauman et al., 2018**; **Thomas et al., 2021**; **Ngo et al., 2019**; **Magrì et al., 2020**; **Kuhn et al., 2018**; **Cerullo et al., 2020**; **Cheng, 2020**; **Pecoraro et al., 2019**). For instance, the studies of VC oral supplementation related to preventing 'common cold' or enhancing immune responses have been highly controversial, often confounded by different administrative routes and quantitative methods (**Padayatty et al., 2004**; **Allan and Arroll, 2014**; **Douglas and Hemilä, 2005**; **Lykkesfeldt and Tveden-Nyborg, 2019**). The results could potentially be confounded by factors including the pre-existing VC levels in the participants (**Prinz et al., 1977**; **Vallance, 1977**; **Anderson et al., 1980**; **Goodwin and Garry, 1983**); genetic variants in the specific VC transporters (**Cahill and El-Sohemy, 2010**); the ill-defined viral pathogens for 'common cold'; and the evolution of viruses to infect humans regardless of VC status. Notably, excess oral supplementation does not increase the steady-state VC concentration beyond the physiological level maintained by absorption and excretion (**Padayatty et al., 2004**). Recently, VC has been injected intravenously to achieve a supraphysiological concentration for treating sepsis and cancers (**Magrì et al., 2020**; **Kuhn et al., 2018**; **Klimant et al., 2018**). The outcomes from these studies were ambiguous as to whether or not the 'mega dose' VC was effective in treating these diseases (**Nauman et al., 2018**; **Thomas et al., 2021**; **Ngo et al., 2019**; **Magrì et al., 2020**; **Klimant et al., 2018**; **Fowler et al., 2019**; **Marik et al., 2017**). Nonetheless, oral and intravenous VC supplementation may selectively benefit the population with VC deficiency or insufficiency that may cause inferior outcomes (**Consoli et al., 2020**; **Fisher et al., 2014**; **Wu et al., 2004**).

VC deficiency has been associated with dysregulated immune responses based on historical observation and experimental evidence. For instance, using *Gulo*-knockout mice that are unable to synthesize VC, studies have shown that VC deficiency impairs the immune response to influenza infection. One study showed that influenza infection in these mice increased the production of pro-inflammatory cytokines (TNF-α, IL-1β) and promoted lung pathology without affecting viral clearance (**Li et al., 2006**). Another study showed that *Gulo*-deficient mice are defective in viral clearance due to decreased type I interferon response (**Kim et al., 2013**). In addition to *Gulo*-deficient mice, guinea pigs are naturally unable to synthesize VC and have been used as a model to show that VC is important for T-cell-dependent antibody response after immunization (**Prinz et al., 1980**; **Feigen et al., 1982**). How VC regulates innate and adaptive immune responses remains unclear. In addition to its antioxidative function, VC is a cofactor for several epigenetic enzymes, including ten-eleven-translocation (TET; **Blaschke et al., 2013**; **Minor et al., 2013**; **He et al., 2015**). TET proteins (TET1, TET2, TET3) are critical for DNA demethylation by oxidizing 5-methylcytosine (5mC) into 5-hydroxymethylcytosine (5hmC) and other minor oxidized bases, which are the intermediates for demethylation (**Lio and Rao, 2019**). Besides enhancing neuronal and hematological differentiations in vitro (**Wu and Zhang, 2017**), VC facilitates the differentiation of induced regulatory T cells by potentiating TET-mediated demethylation of the *Foxp3* intronic enhancer CNS2 (**Sasidharan Nair et al., 2016**; **Yue and Rao, 2020**; **Someya et al., 2017**; **Yue et al., 2016**). These findings strongly suggested that one of the physiological functions of VC is to ensure the proper regulation of the epigenome that is required for coordinated immune responses.

B cells are responsible for the humoral immunity against pathogens in part by differentiating into plasma cells, the terminally differentiated cells specialized in secreting antibodies. Plasma cells express the key transcription factor (TF) BLIMP1 (encoded by *Prdm1*) and can be induced by T-cell-derived signals, including CD40 ligand and IL-21. Using a two-step in vitro system that models the T-dependent plasma cell differentiation, we performed a micronutrient analysis and identified VC as a

potent enhancer for plasma cell differentiation from mouse and human B cells. This effect on plasma cells is specific to VC as other antioxidants had no significant effect. We identified that early B cell activation (first step) is the critical period requiring the presence of VC, which enables the B cells to become plasma cells after IL-21 stimulation. Intriguingly, VC treatment had only minimal influence on the transcriptome and IL-21 proximal signaling, suggesting the effect of VC is likely via the epigenome. Indeed, VC treatment increases the activity of TET as indicated by the elevated 5hmC level. Genome-wide 5hmC profiling revealed several VC-responsive elements at the *Prdm1* locus that the DNA modification statuses were sensitive to VC. Furthermore, we identified that VC augments the association of STAT3 to E27, a critical *Prdm1* downstream enhancer, by facilitating TET2/3-dependent DNA demethylation. Our results suggest that an adequate level of VC is essential for proper antibody response and highlight the influence of micronutrients on cell differentiation via epigenetic enzymes.

## Results

### Vitamin C as an enhancer for plasma cell differentiation

To study the role of micronutrients in plasma cell differentiation, we used a well-characterized B cell culture system (*Nojima et al., 2011*). Mouse naïve B cells were cultured with 40LB, a stromal cell expressing CD40L and BAFF to activate and promote the survival of B cells (*Figure 1A*). Naïve B cells were seeded with 40LB in the presence of IL-4 for 4 days (first step), followed by a secondary co-culture with IL-21 for 3 days to induce the differentiation of plasma cells (second step) distinguished by the surface marker CD138 (Syndecan-1). However, using this two-step culture, only around 20% plasma cells were generated after 7 days (*Figure 1B*).

We reasoned that the relatively inefficient plasma cell differentiation might be due to the paucity of micronutrients, the deficiencies of which have been linked to increased susceptibility to infections (*Maggini et al., 2018*). To determine whether effective B cell differentiation requires micronutrients, we included B27 culture supplement and/or VC to the B cell culture. B27 supplement contains hormones, antioxidants (e.g., vitamin E and reduced glutathione), and other micronutrients (e.g., vitamins A, B7/biotin, selenium). In neurons, the combination of B27 and VC was effective in promoting their differentiation (*Brewer et al., 1993*; *Kim et al., 2018*; *Deng et al., 2015*; *Zhang and Zhang, 2010*). In our B cell culture system, the combination of B27 and VC significantly increased the percentage of plasma cells by at least threefold after the 7-day culture (*Figure 1C and D*). Remarkably, VC was the major component responsible for the enhancement of plasma cell differentiation (*Figure 1C and D*), while B27 alone or B27 without antioxidants (B27-AO) had no effect. These culture supplements and VC had no significant effect on cell numbers and cell death (*Figure 1—figure supplement 1A–C*), suggesting that the increased plasma cells were not due to the preferential increased in survival.

Given that the primary activity of VC relies on its redox function, it is possible that VC relieves oxidative stress that may inhibit plasma cell differentiation. To address this possibility, we cultured B cells in the presence of another antioxidant N-acetylcysteine (NAC). Unlike VC, NAC had no significant effect on plasma cell differentiation (*Figure 1C and D*). In addition, B27 and the culture media also contained other antioxidants (i.e., 2-mercaptoethanol) but were unable to promote plasma cell differentiation (*Figure 1C and D*). The specific activity of VC likely requires an enediol group as the structurally similar stereoisomer erythorbic acid (EA) could similarly promote plasma cell differentiation (*Figure 1—figure supplement 2*). Importantly, the depletion of VC with ascorbate oxidase abolished the plasma-cell-enhancing activity (*Figure 1—figure supplement 3*). Therefore, these data suggest that the effect of VC on plasma cell is specific and may not generalize to other types of antioxidants.

Next, we analyzed the oxidative state in the cultured B cells using a fluorescent dye (CellROX). The data showed that the mock and VC-treated cells similarly had low oxidative levels (*Figure 1E*, left panel), which is supported by the fact that the addition of NAC during the CellROX assay did not significantly decrease the fluorescent signal (*Figure 1E*, second panel). To confirm the validity of the assay, tert-butyl hydroperoxide (TBHP) was added to induce reactive oxygen species (ROS) and increased fluorescent signal (*Figure 1E*, third panel). The presence of VC in the culture media was sufficient to quench the exogenous ROS that could be further removed with the addition of NAC (*Figure 1E*, third and fourth panels). The results demonstrated that the activated B cells in our system exhibited low oxidative stress. Thus, the effect of VC on plasma cells is independent of its well-established general antioxidative function.

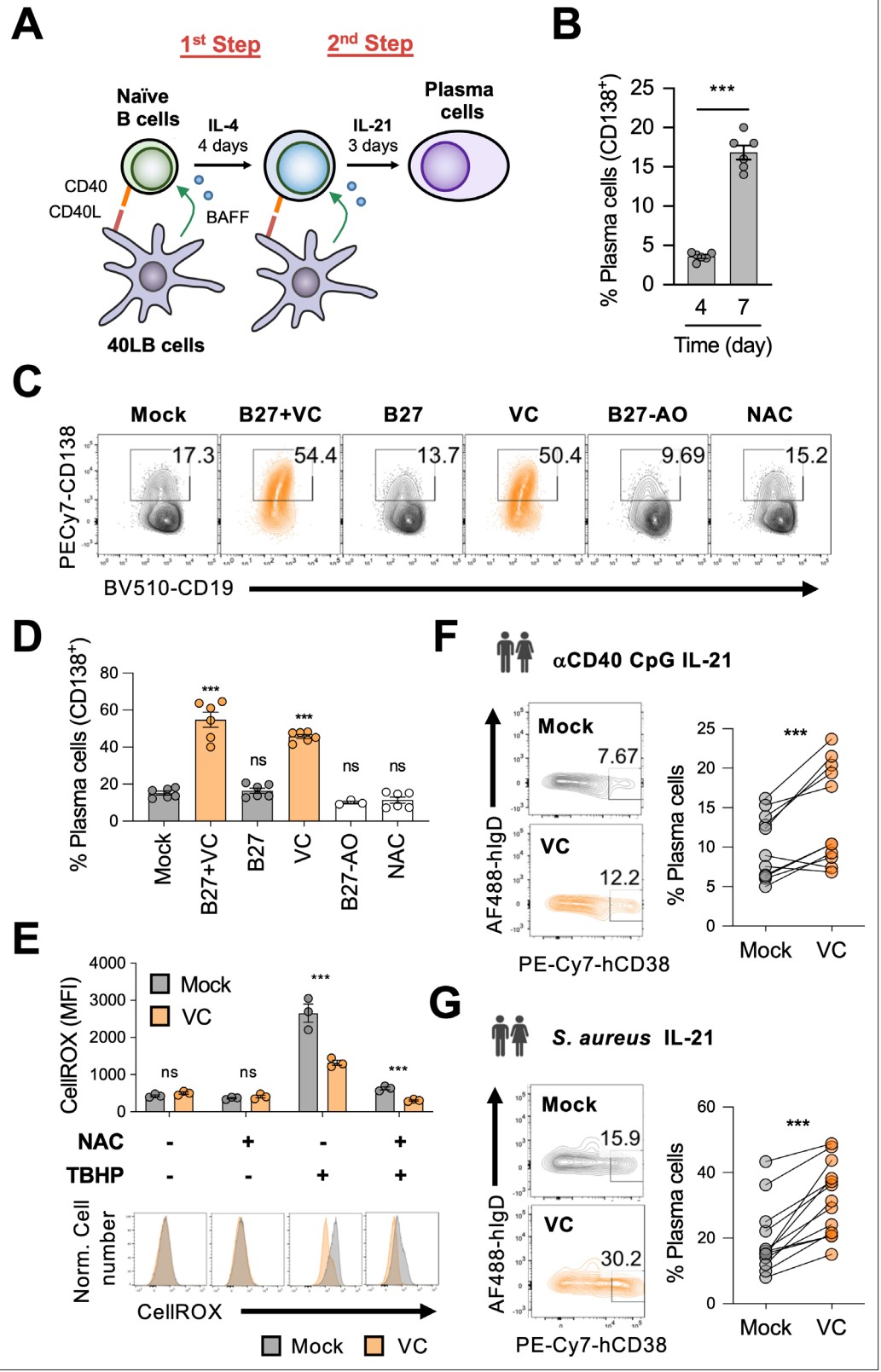

**Figure 1.** Vitamin C (VC) augments mouse and human plasma cell differentiation. (**A**) Schematic depiction of the two-step 40LB cell culture system. Splenic naïve B cells were isolated and cultured with IL-4 and irradiated 40LB cells, which express CD40L and BAFF. Four days after culture, cells were subcultured to new layer of 40LB cells. IL-21 was used to induce plasma cell differentiation. (**B**) Fluorescence-activated cell sorting (FACS) analysis of plasma

*Figure 1 continued on next page*

*Figure 1 continued*

cells generated after each step during the two-step culture. The plasma cell marker CD138 was analyzed on days 4 and 7 (n = 6). (**C, D**) Micronutrient analysis for plasma cell differentiation. B cells were cultured with supplements as indicated and were analyzed by FACS on day 7 (n = 6). (**E**) Low basal oxidative states in cultured B cells. B cells were cultured for 4 days with or without VC and the reactive oxygen species were monitored using a FACS-based fluorescent assay (CellROX). The antioxidant N-acetylcysteine (NAC) and oxidant tert-butyl hydroperoxide (TBHP) were added 30 and 15 min, respectively, prior to the assay as controls (n = 3). (**F, G**) VC enhances the differentiation of human plasma cells. Naïve B cells were isolated from human peripheral blood mononuclear cells (PBMCs) and stimulated for 6 days as indicated to induce plasma cell differentiation in the presence or absence of VC. Data are from 5 to 6 independent experiments with 5–6 donors. All data are from at least two independent experiments. Mean ± SEM is shown for bar charts. Statistical significance was determined by unpaired (**B, E**) or paired (**F, G**) Student's *t*-test and one-way ANOVA with Dunnett's post hoc test (**D**). ***$p < 0.001$. ns, not significant.

The online version of this article includes the following figure supplement(s) for figure 1:

**Figure supplement 1.** Minimal effect of vitamin C (VC) on B cell survival.

**Figure supplement 2.** Vitamin C (VC) stereoisomer and derivative could promote plasma cell differentiation.

**Figure supplement 3.** Ascorbate oxidase (AAO) inhibits vitamin C (VC)-mediated plasma cell differentiation.

**Figure supplement 4.** Vitamin C (VC) enhances plasma cell differentiation and antibody secretions from lipopolysaccharide (LPS)-stimulated B cells.

**Figure supplement 5.** Vitamin C (VC) increases the differentiation of antibody-secreting cells (ASC) from human B cells.

---

The effect of VC is not limited to Th2-associated antibody isotypes (IgG1 and IgE) that are dominant in the 40LB system. VC treatment increased the percentage of plasma cells (*Figure 1—figure supplement 4A*) and the levels of secreted IgG2c and IgG3 in B cells stimulated with lipopolysaccharide (LPS) and LPS with IFN-γ (*Figure 1—figure supplement 4B*). Consistent with the 40LB system, VC also increases the secretion of IgG1 in B cells stimulated with LPS and IL-4 (*Figure 1—figure supplement 4B*). Together, these results suggested that VC promotes plasma cell differentiation and antibody secretion in various stimulated conditions independent of isotypes.

To address whether the activity of VC on plasma cells is conserved in humans, we isolated naïve B cells (*Figure 1—figure supplement 5A*) from the peripheral blood of healthy donors and induced the differentiation of antibody-secreting cells (plasma cell lineage) using two conditions: (1) anti-CD40, Toll-like receptor ligand CpG (ODN), and IL-21; (2) *Staphylococcus aureus* Cowan I (SAC) and IL-21 (*Ettinger et al., 2005*). In addition to stimulating Toll-like receptors, SAC also triggers B cell receptor (BCR) signaling via crosslinking BCR by the high levels of protein A on the bacterial surface (*Romagnani et al., 1981*). Consistent with its effect on mouse cells, VC facilitated the differentiation of human antibody-secreting cells in both conditions (*Figure 1F and G*). To further mimic the in vivo activation of B cells, we stimulated the cells using another two-step system. First, B cells were stimulated with anti-IgM/G, anti-CD40, CpG, and IL-2 for 4 days, followed by IL-2, IL-10, and IL-4 for 3 days (*Hipp et al., 2017*). Under this condition, VC increased the percentage of antibody-secreting cells (IgD^lo CD38^hi) and early plasma cells (CD27^hi CD38^hi; *Figure 1—figure supplement 5B and C*). The antibody-secreting cells cultured with VC downregulated the surface expression of CD19, a characteristic of mature plasma cells (*Figure 1—figure supplement 5B and C*). Together, the results suggested that VC is an important micronutrient that potentiates plasma cell differentiation in both mice and humans.

## VC supplement facilitates the differentiation of bona fide plasma cells

Both naïve and memory B cells are capable of differentiating into plasma cells. While the purified B cells are mainly naïve B cells, it is possible the plasma cells may be derived from the few co-purified memory B cells. To test that VC promotes plasma cell differentiation from naïve B cells, we isolated B cells from IgHCGG mice, a BCR knock-in strain in which most B cells express BCR specific for a foreign protein (chicken gamma globulin) and have a naïve phenotype (*Jacobsen et al., 2018*). The data showed that VC has similar effect on the B cells isolated from WT and IgHCGG mice (*Figure 2—figure supplement 1*), suggesting that the plasma cells generated in the VC culture are mostly derived from naïve B cells.

Our data revealed that VC promotes plasma cell differentiation using surface expression of CD138 as the marker. To confirm the identity of these cells, we analyzed the protein expression of BLIMP1

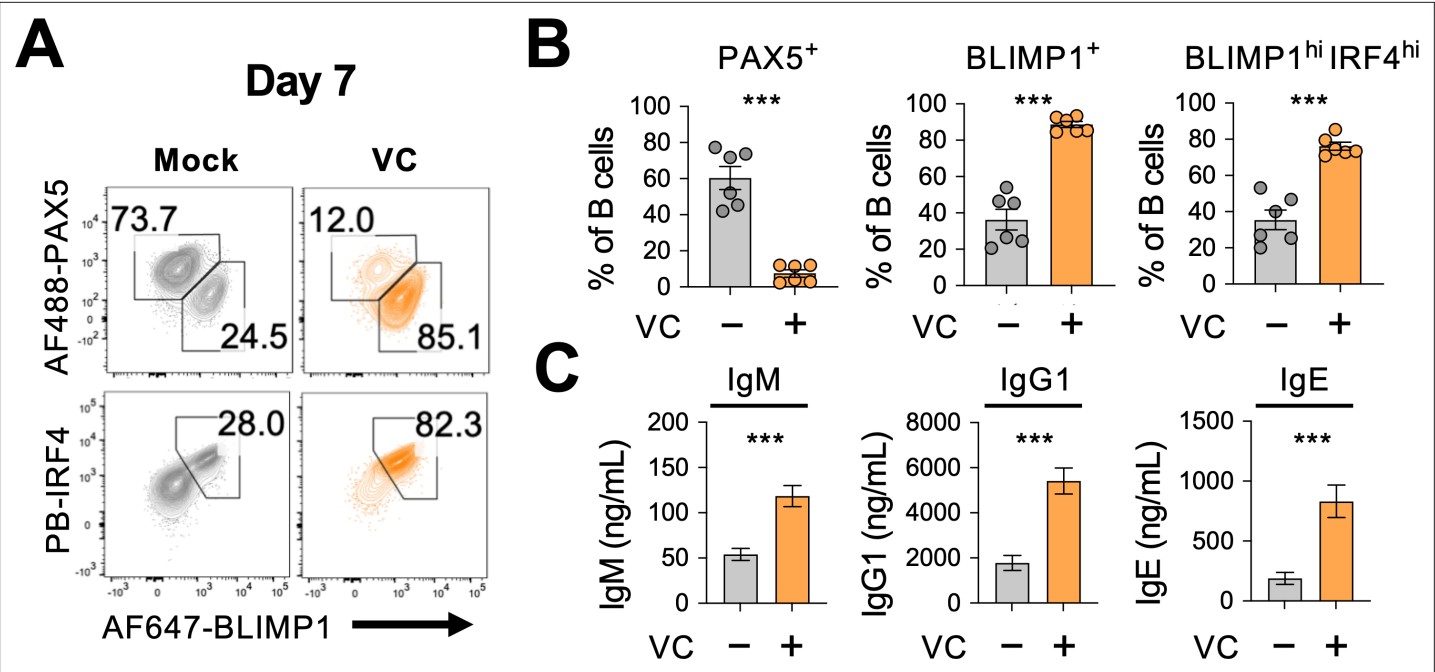

**Figure 2.** Vitamin C (VC) promotes the generation of bona fide plasma cells. (**A, B**) Expression analysis of lineage transcription factors (TFs). B cells were cultured as indicated for 7 days, and the expression of transcription factors was analyzed by intracellular staining and fluorescence-activated cell sorting (FACS) (n = 6). Representative FACS plots (**A**) and summaries (**B**) are shown. Note that BLIMP1 and IRF4 are important TF plasma cell differentiation. (**C**) Antibody secretion by VC-induced plasma cells. Culture supernatants were collected after day 7, and the secreted antibodies with indicated isotypes (IgM, IgG1, IgE) were analyzed by ELISA (n = 6). All data are from at least two independent experiments. Mean ± SEM is shown for bar charts, and the statistical significance was determined by unpaired Student's *t*-test. \*\*\*p<0.001.

The online version of this article includes the following figure supplement(s) for figure 2:

**Figure supplement 1.** Vitamin C (VC) similarly promotes plasma cell differentiation from polyclonal and monoclonal naïve B cells.

**Figure supplement 2.** Minimal antibody secretion and differentiation of plasma cells at early stage.

**Figure supplement 3.** Vitamin C (VC) increases plasma cells with a mature phenotype in an extended culture.

and IRF4, two TFs important for the plasma cell lineage. Consistent with CD138, the percentage of BLIMP1- and IRF4-positive cells increased substantially after 7-day culture with VC (*Figure 2A and B*). Similar to mature plasma cells, VC-treated B cells also downregulated the B-lineage TF PAX5 (*Figure 2A, upper panel, and B*). While B cells secreted minimal number of antibodies after the first-step culture regardless of VC (*Figure 2—figure supplement 2*), VC treatment significantly enhanced the antibody secretion after IL-21 stimulation during the second-step culture (*Figure 2C*). The effect of VC was not simply due to accelerating the kinetics of plasma cell differentiation as the VC treatment group still had a significantly higher proportion (~84%) of plasma cells compared to control (~35%) in an extended culture of B cells with 40LB (*Figure 2—figure supplement 3A and B*). Similar to mature plasma cells, the VC-treated B cells significantly downregulated CD19 (*Figure 2—figure supplement 3C*). Together, these results demonstrated that VC enhanced the differentiation of bona fide plasma cells.

## The initial activation step of plasma cell culture is the critical period for VC

As described above (*Figure 1A*), in vitro plasma cell differentiation is a two-step process, where naïve B cells are initially activated with IL-4 (first step) then with IL-21 to induce plasma cell differentiation (second step). To identify the critical period for VC, we treated the cells with VC at different time points as indicated (*Figure 3A*) and analyzed the cells by fluorescence-activated cell sorting (FACS) on day 7 for CD138 expression. Consistent with the above result, the addition of VC throughout the culture significantly enhanced plasma cell differentiation compared to control (*Figure 3A*; compare I vs. IV). The data showed that the addition of VC during the first step is sufficient to enhance plasma

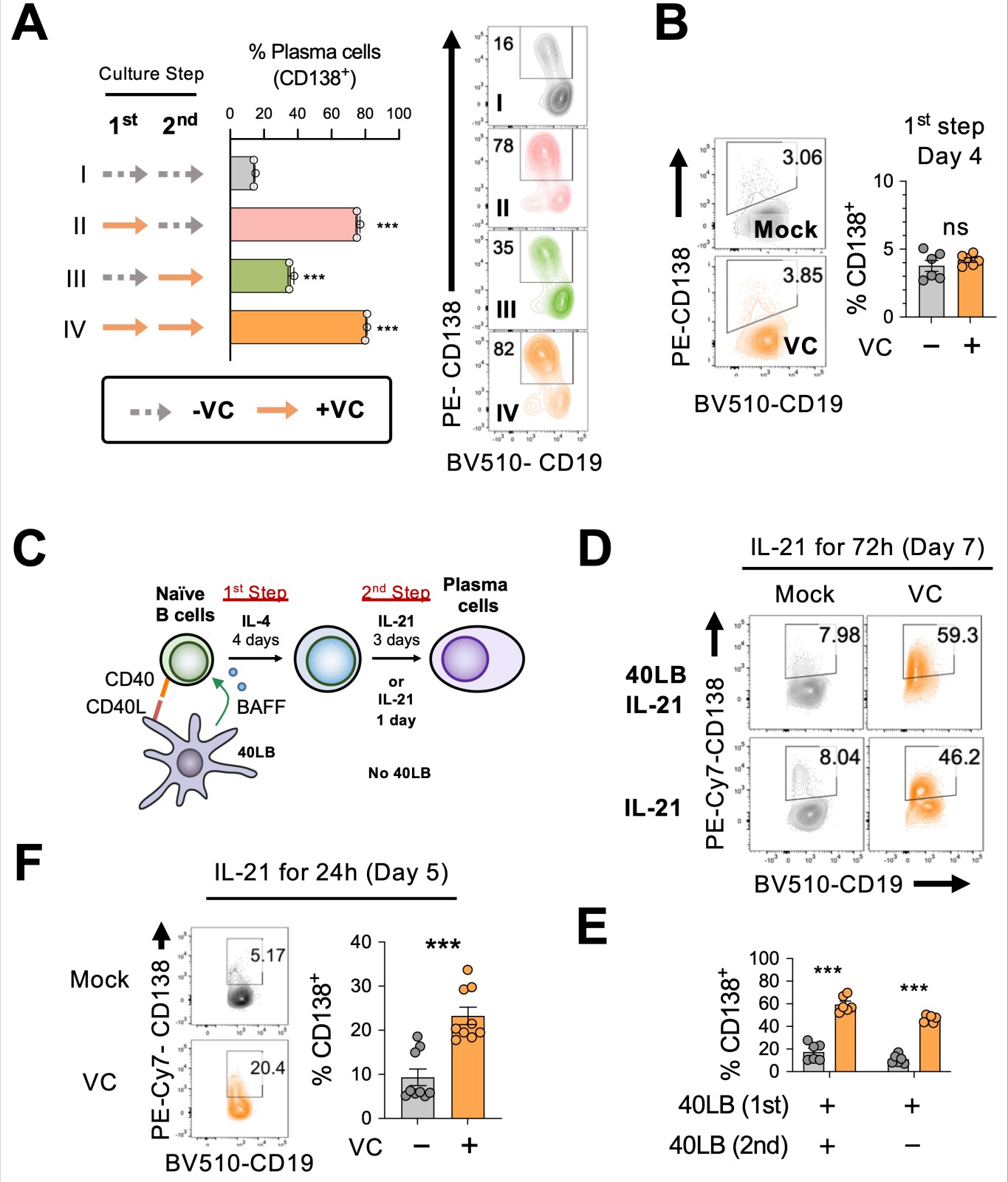

**Figure 3.** Vitamin C (VC) potentiates the IL-21-dependent differentiation of plasma cell during the initial activation phase. (**A**) VC is required during activation phase. B cells were treated with or without VC as indicated for the first- and second-step culture. Dotted arrow represents control and solid orange arrow indicates VC. Plasma cell differentiation (%CD138⁺) was analyzed on day 7 by fluorescence-activated cell sorting (FACS). Representative FACS plots are shown on right and summarized data are shown on left. Roman numbers are used for correlation between panels. (**B**) VC has no

*Figure 3 continued on next page*

*Figure 3 continued*

significant effect on the percentage of plasma cells after first-step culture. Representative FACS plots for plasma cell marker CD138 of B cells on day 4 (left), and the summarized data (right) are shown (n = 6). (**C–E**) IL-21 alone is sufficient to induce plasma cell differentiation during the second step. (**C**) Schematic representation of the B cell culture used in the next panels. B cells were cultured with IL-21 for indicated time without the 40LB stromal cells. (**D, E**) Cells were either cultured with or without 40LB in the presence of IL-21 during the second step. Percentage of plasma cells (%CD138$^+$) was analyzed by FACS on day 7 (n = 6). The representative FACS plots (**D**) and summarized data (**E**) are shown. (**F**) IL-21 stimulation induced plasma cell differentiation after 24 hr. B cells were cultured in the absence (mock) or presence (VC) of VC during the first step with IL-4 and 40LB. Cells were stimulated with IL-21 alone for 24 hr and the percentage of CD138$^+$ cells was analyzed by FACS (n = 9). All data are from at least two independent experiments. Mean ± SEM is shown for bar charts, and the statistical significance was determined by one-way ANOVA with Dunnett's post hoc test (**A**) and unpaired Student's *t*-test (**B, E, F**). \*\*\*p<0.001. ns, not significant.

The online version of this article includes the following figure supplement(s) for figure 3:

**Figure supplement 1.** IL-21 alone during later phase is sufficient for plasma cell differentiation.

cell differentiation (***Figure 3AII***), while the effect is less pronounced when VC is added during the second step (***Figure 3AIII***). The higher percentage of plasma cells after VC treatment is not due to accelerated kinetics of plasma cell differentiation. B cells were similar in function and phenotype after the initial 4 days of culture regardless of VC treatment (***Figure 3B***, ***Figure 2—figure supplement 2***). These data demonstrated that VC is important for conditioning the B cells during the initial activation period.

During the second-step culture, B cells are stimulated with 40LB and IL-21 to induce plasma cell differentiation. To determine whether VC facilitates plasma cell differentiation required continuous signals from CD40L and BAFF, we cultured the B cells with IL-21 in the presence or absence of 40LB cells during the second step as indicated (***Figure 3C***). The results show that VC significantly enhanced plasma cell differentiation regardless of 40LB cells by measuring CD138 (***Figure 3D and E***), antibody secretion (***Figure 3—figure supplement 1A***), and TF expression (***Figure 3—figure supplement 1B***). While the percentage of plasma cells was similarly low in mock- and VC-treated cells after first-step culture (***Figure 3B***), stimulation with IL-21 for 24 hr was sufficient to induce significantly more plasma cells from the VC-treated B cells (***Figure 3F***). These results suggest that VC conditions B cells to be responsive to the IL-21-induced plasma cell differentiation.

## VC has limited effect on the proximal IL-21 signaling and transcriptome

IL-21 is produced by T cells and induces the phosphorylation of STAT3 after binding to IL-21 receptor (IL-21R) on B cells. To address whether VC promotes plasma cell differentiation by altering IL-21

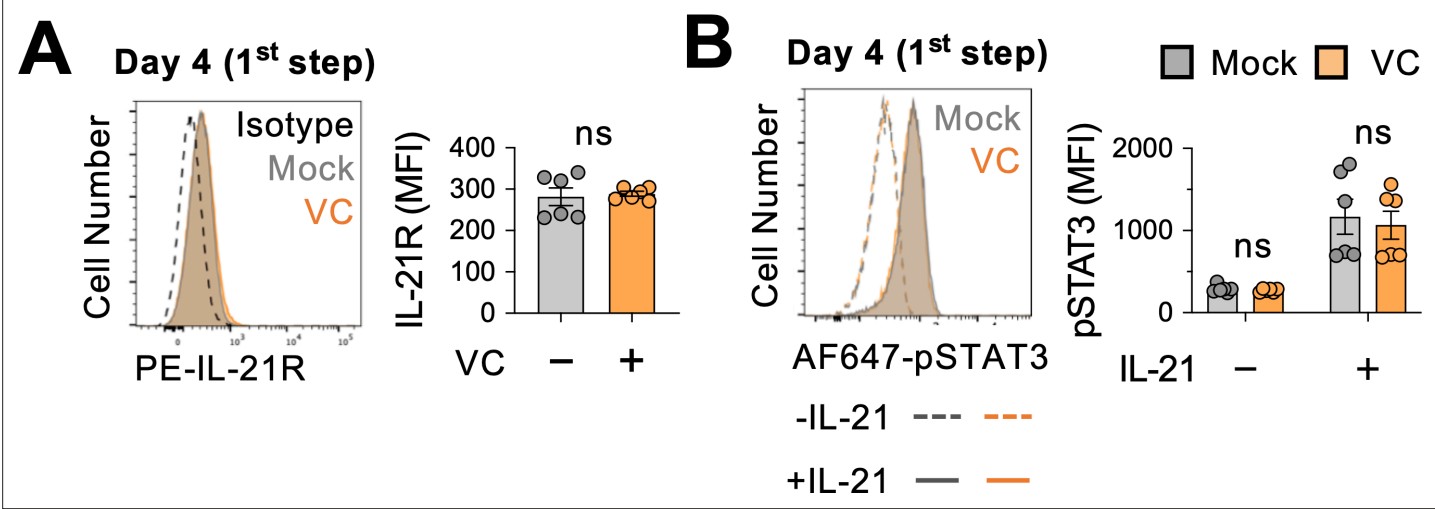

**Figure 4.** Vitamin C (VC) did not promote the proximal IL-21 signaling. (**A**) VC has no effect on IL-21 receptor expression. Mock or VC-treated B cells were cultured for 4 days, and the expression of IL-21R was analyzed by fluorescence-activated cell sorting (FACS) (n = 6). (**B**) VC does not alter the IL-21-induced STAT3 phosphorylation. B cells from day 4 culture were stimulated with IL-21 for 30 min, and the level of STAT3 phosphorylation (pSTAT3) was analyzed by FACS (n = 6). Representative histograms are shown on left and summarized data on right. All data are from at least two independent experiments. Mean ± SEM is shown for bar charts, and the statistical significance was determined by unpaired Student's *t*-test. ns, not significant.

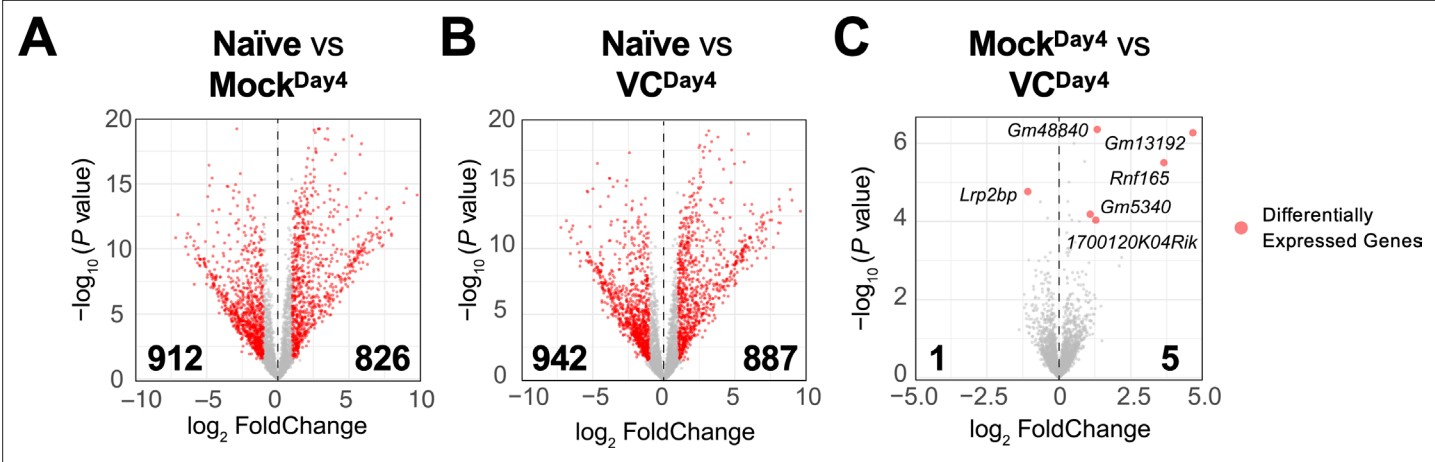

**Figure 5.** Limited effect of vitamin C (VC) on the transcriptome in activated B cells. The mRNA was isolated from naïve and indicated day 4 activated B cells, and the transcriptomes were analyzed by RNA-seq (see 'Materials and methods'). Comparisons shown are between (**A**) naïve vs. mock, (**B**) naïve vs. VC, and (**C**) mock vs. VC. Three independent biological replicates were used for each group. Red dots indicate differentially expressed genes with an adjusted p-value≤0.01 and log$_2$ fold change ≥ 1. Numbers indicate the numbers of differentially expressed genes. Gene names for all differentially expressed genes are shown in (**C**).

signaling, we cultured the B cells with 40LB and IL-4 for 4 days, and the expression of IL-21R was analyzed by flow cytometry. The data showed that IL-21R expression was comparable between mock- and VC-treated cells (*Figure 4A*). To test whether VC may enhance the signaling capacity of IL-21R, we stimulated the cells on day 4 with IL-21 for 30 min and analyzed STAT3 Tyr705 phosphorylation (pSTAT3). The results showed that VC has no significant effect on the STAT3 phosphorylation in response to IL-21 (*Figure 4B*). Note that the same concentration of IL-21 was used for both the short-term stimulation (30 min) and the second-step culture. These data suggest that VC-mediated conditioning of the B cells is not due to a heightened IL-21 proximal signaling.

Based on the above results, VC poises B cells toward plasma cell lineage without affecting the overall phenotypes (*Figure 3B*, *Figure 2—figure supplement 2*) and IL-21 signaling (*Figure 4*) after initial activation. To identify the potential genes regulated by VC, we analyzed the transcriptomes of naïve and day 4 activated B cells in the presence or absence of VC by RNA-seq. As expected, B cell activation induced substantial changes in transcriptomes, with thousands of differentially expressed genes (DEGs; *Figure 5A and B*). Surprisingly, the transcriptomes of B cells are highly similar between control and VC-treated cells (*Figure 5C*), with only one and five genes differentially decreased and increased, respectively. Among the DEGs, the majority of them are non-coding transcripts (e.g., *Gm48840, Gm13192, Gm5340*) and proteins with unknown function (1700120K04Rik, *Lrp2bp*). One of the genes induced by VC is Rnf165/Ark2C, a ubiquitin E3 ligase that has been associated with impaired motor neuron axon growth (*Kelly et al., 2013*) with unknown function in the immune system. Nonetheless, based on the minimal number of DEGs, we speculated that VC might condition the B cells without affecting the transcriptome.

## The effect of VC is dependent on either TET2 or TET3

In addition to being an antioxidant that scavenges ROS, the redox activity of VC also reduces Fe(III) to Fe(II) to maintain the activity of epigenetic enzymes, including TET family proteins (TET1, TET2, TET3). TET proteins are essential for DNA demethylation by oxidizing 5mC to mainly 5hmC (*Lio and Rao, 2019*; *Wu and Zhang, 2017*; *Tsioupis et al., 2020*). Previous studies have shown that TET2 and TET3 are required for B cell development and function (*Tanaka et al., 2020*; *Stremenova Spegarova et al., 2020*; *Schoeler et al., 2019*; *Lio et al., 2019*; *Dominguez et al., 2018*; *Orlanski et al., 2016*). Therefore, we hypothesize that VC may affect the B cell epigenome by enhancing the enzymatic activity of TET. We isolated B cells from control (*Cd19$^{Cre/+}$*), *Tet2*-deficient (*Cd19$^{Cre/+}$ Tet2$^{fl/fl}$*), or *Tet3*-deficient (*Cd19$^{Cre/+}$ Tet3$^{fl/fl}$*) mice and cultured them in the presence or absence of VC. While VC greatly increased the percentage of plasma cells generated in control B cells (*Figure 6A–D*; WT), the differentiation was slightly impaired in the *Tet2*- or *Tet3*-deficient cells (*Figure 6A–D*). As TET2 and TET3

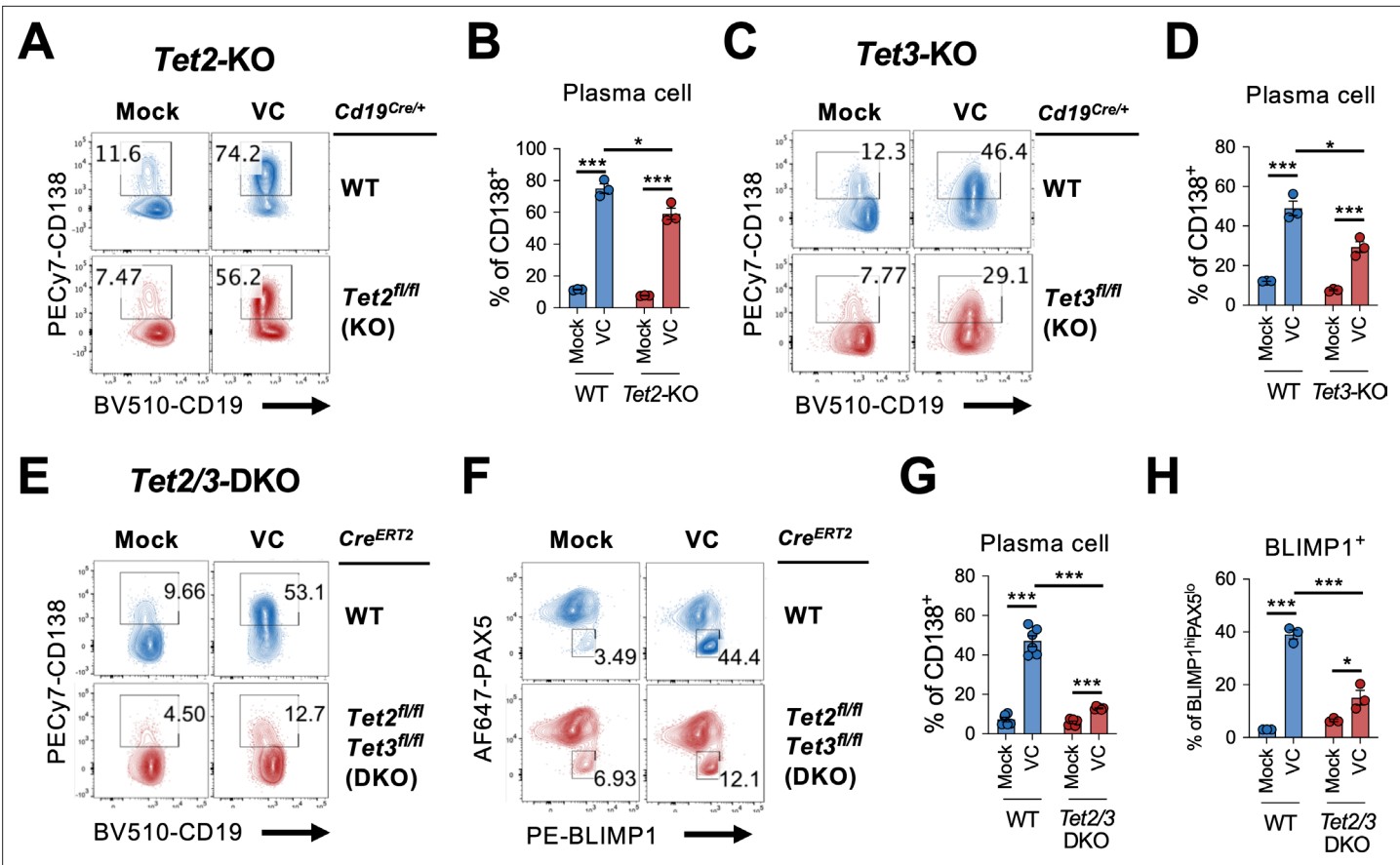

**Figure 6.** TET2 or TET3 is required for vitamin C (VC)-mediated enhancement of plasma cell differentiation. (**A–D**)*Tet2* or *Tet3* is sufficient for VC-mediated increase in plasma cells. (**A, B**) B cells from WT (*Cd19^Cre/+*) or *Tet2* conditional-deficient (*Cd19^Cre/+ Tet2^fl/fl*; *Tet2*-KO) mice isolated and cultured as in *Figure 1A* with or without VC. Percentage of CD138+ cells was analyzed by fluorescence-activated cell sorting (FACS) on day 7 (n = 3). (**C, D**) B cells were isolated from either WT (*Cd19^Cre/+*) or *Tet3* conditional KO (*Cd19^Cre/+ Tet3^fl/fl*; *Tet3*-KO) and cultured as in (**A**; n = 3). Representative FACS plots (**A, C**) and summarized data (**B, D**) are shown. (**E–H**) *Tet2* or *Tet3* are required for the effect of VC. Control (*Cre^ERT2*) or *Tet2/Tet3* conditional-deficient (*Cre^ERT2 Tet2^fl/fl Tet3^fl/fl*) mice were injected with tamoxifen for five consecutive days and B cells were isolated by cell sorting on day 8. The expression of *Rosa26*-YFP was used as a surrogate marker for Cre activity. B cells were cultured as in (**A**) or (**C**), and the percentage of CD138 (**E**) and intracellular staining of transcription factors (TFs) (**F**) were analyzed by FACS on day 7. Summarized data are shown in (**G**) and (**H**). All data are from at least two independent experiments. Mean ± SEM is shown for bar charts, and the statistical significance was determined by unpaired Student's *t*-test. \*\*\*p<0.001, \*p<0.05. ns, not significant.

often function redundantly, we isolated B cells from *Tet2 and Tet3* conditional double-deficient mice to address whether the effect of VC is mediated by TET enzymes. Since the deletion of *Tet2* and *Tet3* using *Cd19^Cre* resulted in aggressive B cell lymphoma at an early age (unpublished observation), we used a tamoxifen-inducible *Tet2/3*-deletion system to circumvent this caveat as previously described (*Lio et al., 2019*). In this system, *UBC^Cre-ERT2 Tet2^fl/fl Tet3^fl/fl* and control *UBC^Cre-ERT2* mice were injected with tamoxifen for five consecutive days to induce the deletions of *Tet2* and *Tet3*. B cells were isolated on day 8 and cultured as above. Remarkably, the effect of VC on plasma cell differentiation dramatically decreased in the absence of TET2 and TET3 as measured by percentages of CD138+ (*Figure 6E and G*) or BLIMP1+ (*Figure 6F and H*) B cells. Therefore, these results strongly suggested that the ability of VC to enhance plasma cell differentiation is via TET2 and TET3.

## VC remodels the genome-wide 5hmC modification

TET enzymes oxidize 5mC into 5hmC, a stable epigenetic medication and an intermediate for DNA demethylation (*Figure 7—figure supplement 1A*; *Wu and Zhang, 2017*; *Tsiouplis et al., 2020*; *Lio et al., 2020b*). To confirm that VC indeed enhances TET activity, we used DNA dot blot to semi-quantitatively measure the level of 5hmC in B cells. Naïve B cells had the highest density of 5hmC

compared with B cells from day 4 or day 7 culture (**Figure 7A**). The decreased 5hmC is consistent with the passive dilution of 5hmC after cell divisions as the 5hmC modification pattern is not replicated on the newly synthesized DNA (**Figure 7—figure supplement 1B and C**). The addition of VC significantly increased the amount of 5hmC compared to those in the mock control B cells at both time points (**Figure 7A**; compare mock and VC), validating that VC enhances TET activity in B cells. These results imply that while TET may be recruited to the chromatin by transcription factors in the absence of VC, TET may have a lower probability to oxidize 5mC to 5hmC due to the decreased enzymatic activity.

To identify the targeted genomic regions regulated by TET, we analyzed the genome-wide 5hmC distribution HMCP, a CLICK-chemistry-based 5hmC pull-down method modified from a previously published protocol (**Song et al., 2011**). Overall, the number of 5hmC-enriched regions correlated with the levels of 5hmC (**Figure 7A**): naïve B cells contained the most 5hmC-enriched regions; B cells activated for 4 days (Mock$^{Day4}$) had the lowest; B cells cultured with VC (VC$^{Day4}$) had significantly more 5hmC-enriched regions compared to mock control (**Figure 7B**). Analysis of the differentially hydroxymethylated regions (DhmRs) revealed that B cells cultured for 4 days without VC (Mock$^{Day4}$) lost the majority (87.6%; **Figure 7C**, left panel, and **Figure 7—figure supplement 2A**) of the 5hmC-enriched regions that were identified in naïve B cells. Smaller percentages of peaks are either maintained (7.9%) or gained (4.4%) that may represent the loci with stronger TET recruitment and a higher probability of demethylation despite the lack of VC. In B cells cultured with VC (VC$^{Day4}$), considerable numbers of DhmRs have decreased in 5hmC (59.2%; **Figure 7C**, middle panel, and **Figure 7—figure supplement 2B**), suggesting that these regions may undergo passive DNA demethylation. A significant number of regions either maintained (24.3%) or gained 5hmC (16.5%) after culture. Most of these 5hmC-enriched regions were distal to the transcription start sites (**Figure 7—figure supplement 2C–E**), suggesting that many of these regions might be potentially regulatory elements as previously described (**Lio et al., 2019**).

To understand the effect of VC on TET-mediated epigenome remodeling, we focused on the regions between the activated B cells from mock and VC groups. The analysis of DhmRs revealed that VC induced a significant number of 5hmC-enriched regions (72.3%; **Figure 7C**, right panel, and **Figure 7D**), while a small number of regions have decreased 5hmC (3.7%), which may represent the regions undergo demethylation. Motif analysis of the regions enriched in VC-treated cells showed modest enrichment of motifs from TF families, including basic helix-loop-helix (bHLH), nuclear receptor (NR), and zinc fingers (Zf; **Figure 7—figure supplement 3A**). The DhmRs preferentially found in mock control were enriched in bHLH, basic leucine zipper (bZIP), and Zf family TFs (**Figure 7—figure supplement 3B**). How and whether these TFs may collaborate with TET to facilitate DNA modification remains to be addressed. Altogether, the data demonstrated that VC enhanced the enzymatic activity of TET, resulting in a substantial increase in 5hmC throughout the genome.

## TET-responsive regulatory elements at the *Prdm1* locus

*Prdm1*-encoded BLIMP1 is the TF critical for the differentiation and function of plasma cells. The above results showed that VC promotes plasma cell differentiation via TET-mediated deposition of 5hmC and potential demethylation. Therefore, we hypothesize that VC may affect the DNA modification status at the *Prdm1* locus to increase the permissiveness of IL-21/STAT3-mediated upregulation. In naïve B cells, the *Prdm1* gene body and several downstream regions are enriched in 5hmC (**Figure 8A**, top track 'naïve'). In B cells activated without VC (Mock$^{Day4}$), the majority of the regions showed decreased 5hmC with the exception of E27, a previously described enhancer (teal bars at E27 in **Figure 8B**; **Kwon et al., 2009**). As mentioned above, the regions with 5hmC gain in the mock group likely represent stronger TET recruitment (**Figure 8A**, compare naïve vs. Mock$^{Day4}$). VC induced the enrichment of 5hmC in at least 14 locations (teal bars in **Figure 8B**, *Mock$^{Day4}$* vs. *VC$^{Day4}$*). One of the most pronounced enrichment was at E58, a previously undescribed element located at +58 kb relative to the start of *Prdm1* (**Kwon et al., 2009**). Overlaying a previously published DNA methylation data demonstrated that both E27 and E58 are methylated in naïve B cells and are at the center or the edge of demethylating regions, respectively (**Figure 8C**). This observation is consistent with a previous notion that the boundaries between methylated and demethylation regions are enriched in 5hmC (**Han et al., 2016**).

Since VC induced TET-mediated 5hmC modifications at the *Prdm1* locus, we speculate that the DNA modification status may affect STAT3 association with DNA. To test this hypothesis, we analyzed

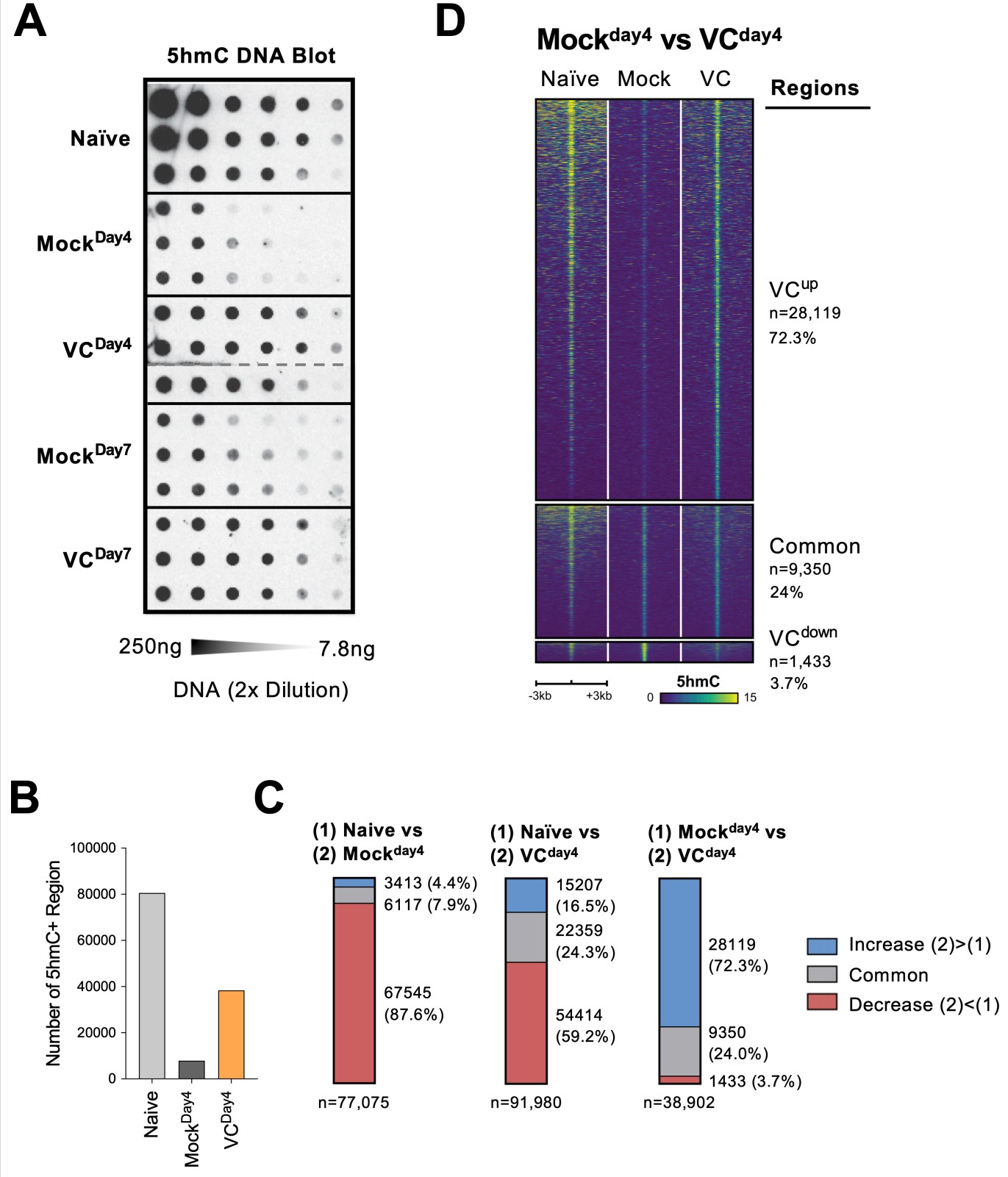

**Figure 7.** Vitamin C (VC) induced a global increase in 5hmC modification. (**A**) VC increased total 5hmC in B cells. DNA from indicated B cells were isolated and the total 5hmC levels were analyzed using cytosine 5-methylenesulfonate (CMS) dot blot ('Materials and methods'). Note that CMS is the immunogenic product of 5hmC after bisulfite treatment. DNA was serially twofold diluted starting from 250 ng to ~7.8 ng. Three biological replicates were blotted for each group. (**B–D**) The genome-wide 5hmC enrichment was analyzed using a CLICK-chemistry-based pull-down method (HMCP).

*Figure 7 continued on next page*

*Figure 7 continued*

Two biological replicates were used for each condition (naïve, Mock$^{Day4}$, VC$^{Day4}$; see 'Materials and methods' for details). (**B**) VC treatment increased the numbers of 5hmC-enriched region. The number of 5hmC-enriched (5hmC+) regions was called using HOMER, and the numbers of consensus regions between replicates are plotted. (**C**) VC maintains pre-existing and induces de novo 5hmC-enriched regions. Pairwise comparisons between the three groups are shown. Numbers and percentages of regions are shown for each comparison. Colors indicate the 5hmC status of the regions. Number of 5hmC+ region in naïve B cells and B cells with or without VC treatment. (**D**) Visualization of 5hmC at differential regions between B cells from Mock$^{Day4}$ and VC$^{Day4}$ groups. The 5hmC enrichment is plotted as heatmaps around (±3 kb) the differential and common regions between Mock$^{Day4}$ and VC$^{Day4}$ (right panel of **C**). Normalized 5hmC counts are plotted as color scale, and each row represents a region.

The online version of this article includes the following source data and figure supplement(s) for figure 7:

Source data 1. Original blot for *Figure 7A* cytosine 5-methylenesulfonate (CMS) dot blot.

Figure supplement 1. Maintenance DNA methylation and 5hmC-mediated passive DNA demethylation.

Figure supplement 2. Analyses of 5hmC-enriched regions.

Figure supplement 3. Enrichment of transcription factor (TF) motifs at differential 5hmC regions.

the STAT3 binding at *Prdm1* locus in B cells after the first-step culture (*Figure 8D*). Without IL-21 stimulation, only low levels of STAT3 were observed and no significant difference was observed between mock and VC-treated B cells (*Figure 8D*, top two tracks). Strikingly, compared to Mock control, VC strongly increased STAT3 association at the promoter and E27 in response to IL-21 (*Figure 8D*, bottom two tracks). The results show that VC and TET are critical regulators of STAT3 association at E27, in which the STAT3 binding site is essential for IL-21-induced *Prdm1* expression (*Kwon et al., 2009*).

In addition to E27, E58 showed one of the highest proportions of 5hmC after VC treatment. Interestingly, while STAT3 does not associate with E58 in B cells, the analysis of previously published ChIP-seq (*Kwon et al., 2009*) showed that STAT3 could bind to E58 in T cells (*Figure 8—figure supplement 1*; bottom two tracks). Whether the 5hmC at E58 influences *Prdm1* expression remained to be determined.

## VC induced DNA demethylation and facilitated STAT3 binding at E27

From the above results, we identified E27 as a VC-responsive element that regulates *Prdm1* expression. 5hmC analysis of mock-treated B cells showed that E27 potentially had a higher TET recruitment based on the 5hmC enrichment despite the lack of VC (*Figure 8A*, compare naïve vs. Mock$^{Day4}$). The decreased 5hmC in the VC-treated cells may be due to the DNA demethylation that removes 5mC, the substrate for TET-mediated 5hmC generation (*Figure 7—figure supplement 1*). To test whether E27 is a hotspot for TET-mediated DNA modification and demethylation, we analyzed the DNA methylation status using bisulfite amplicon sequencing coupled with Nanopore sequencer. As regular bisulfite sequencing is not able to distinguish 5mC and 5hmC, we collectively refer the modified cytosine as 'methylated' and the majority of the methylated cytosines are 5mC (*Lio et al., 2019*). We found that in response to VC a stretch of CpGs at E27 (differentially methylated region [DMR]) was hypomethylated compared to mock control (*Figure 9A*, *Figure 9—figure supplement 1A*). The loss of 5mC explains the decreased 5hmC level at E27 in VC-treated B cells as 5hmC is derived from 5mC (*Figure 7—figure supplement 1*). Consistent with our hypothesis, E27 DMR remained methylated after VC treatment in the absence of TET2 and TET3 (*Figure 9B*). Interestingly, E58 remained mostly methylated (5mC + 5hmC; *Figure 9—figure supplement 1B*) despite enrichment in 5hmC after VC treatment (*Figure 8A*). As regular bisulfite sequencing cannot distinguish 5mC and 5hmC, the percentage and function of 5hmC at E58 await to be determined. In contrast to E27 and E58, the CpGs at *Prdm1* promoter were mostly demethylated even in the absence of TET2/3 (*Figure 9—figure supplement 1C*), suggesting that the inability to upregulate *Prdm1* is not due to promoter hypermethylation in the absence of VC or TET enzymes.

Previous work has shown that the STAT3 motif at E27 is critical in IL-21-induced *Prdm1* expression (*Kwon et al., 2009*). We found that STAT3 motif is nested within the E27 DMR while the motif is devoid of CpG (*Figure 9A*, *Figure 9—figure supplement 2*). To confirm STAT3-E27 binding is regulated by DNA methylation, we performed ChIP-qPCR using B cells isolated from control or *Tet2/3*-deficient mice. VC strongly increased STAT3 binding to E27 in B cells stimulated with IL-21 (*Figure 9C*). However, the binding dramatically diminished in the absence of TET2/3 (*Figure 9C*). The data strongly

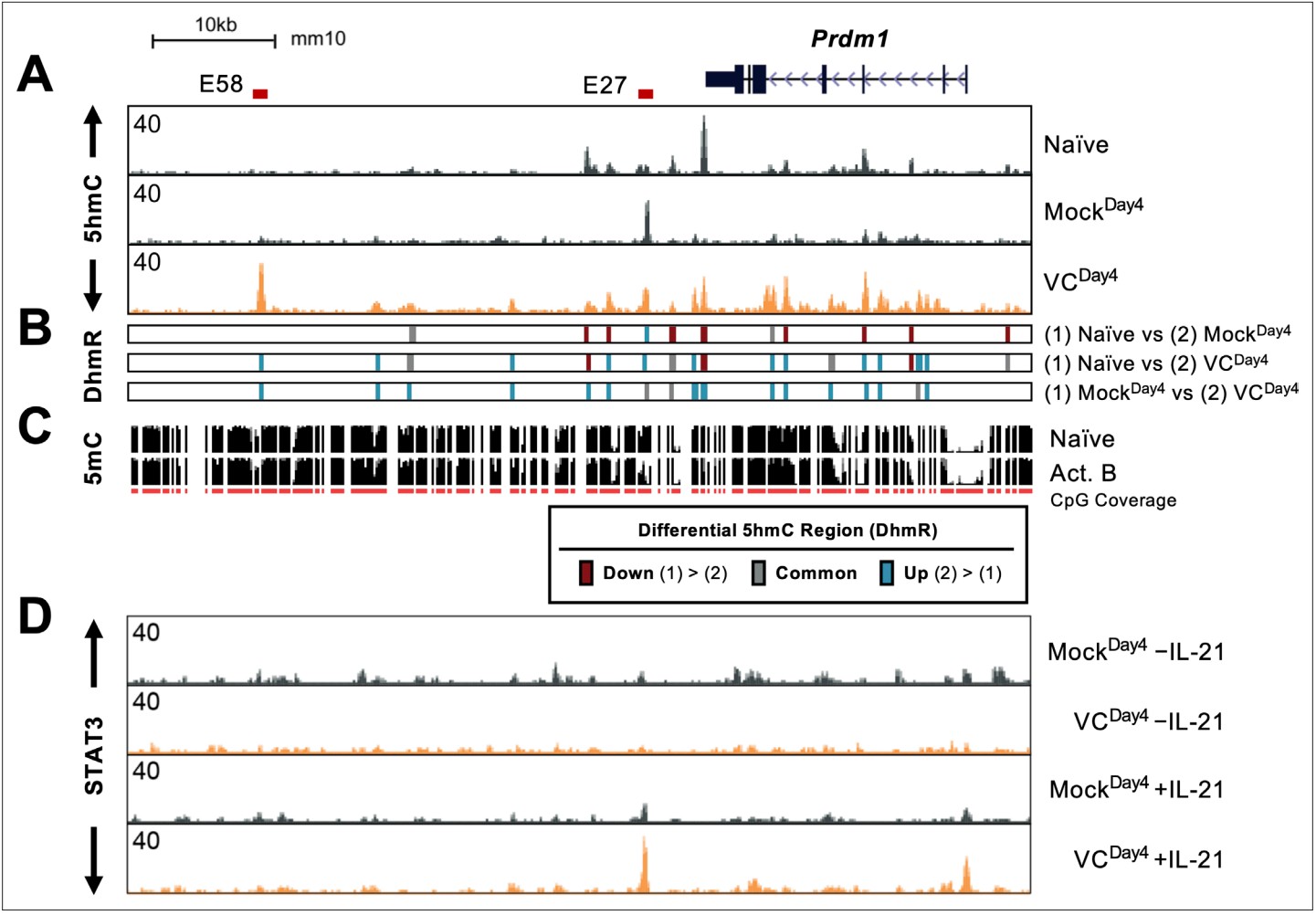

**Figure 8.** Vitamin C (VC) induced 5hmC modification and enhanced STAT3 binding at the *Prdm1* locus. (**A**) Genome browser tracks of the 5hmC enrichment at the *Prdm1* locus (mm10 chr10:44,390,000–44,464,000). Data shown are the average of two biological replicates. The locations of E27 and E58 are shown above. (**B**) Differential 5hmC-enriched regions (DhmRs) between the indicated group as in *Figure 7C*. Colors indicate the differential status of the regions as depicted in the legend. (**C**) DNA methylation in naïve and 48 hr-activated B cells from previous publication (*Lio et al., 2019*; *Kieffer-Kwon et al., 2013*). The height of the black bars indicates the percentage of CpG methylation, and the red track (CpG) indicates the CpGs that were covered in the analysis. (**D**) VC enhanced STAT3 association at E27. B cells were cultured with or without VC for 4 days as in *Figure 4* and treated with or without rmIL-21 (10 ng/mL) for 6 hr. STAT3 binding was analyzed by ChIP-seq. The signals on the genome browser tracks are the average from two biological replicates.

The online version of this article includes the following figure supplement(s) for figure 8:

**Figure supplement 1.** IL-21 induces STAT3 binding at E58 in T cells.

suggest that VC facilitates plasma cell differentiation via TET-mediated demethylation of E27, which then becomes permissive to the binding of STAT3 and augments *Prdm1* transcription.

In conclusion, our data showed that plasma cell differentiation is sensitive to the level of VC, which can affect the epigenome by modulating the activity of epigenetic enzymes including TET. Heightened TET activity may increase the probability of the DNA modification/demethylation at critical regulatory elements and may skew the lineage decision of the differentiating B cells (*Figure 10*).

## Discussion

VC has long been known for its broad effect on immune responses. Long-term VC deficiency results in *scurvy*, a disease historically associated with increased susceptibility to infections (*Baron, 2009*; *Kuhn et al., 2018*; *Hemilä, 2017*). While recent studies have focused on VC supplements for disease treatments or prevention (*Nauman et al., 2018*; *Ngo et al., 2019*; *Magrì et al., 2020*; *Kuhn et al.,*

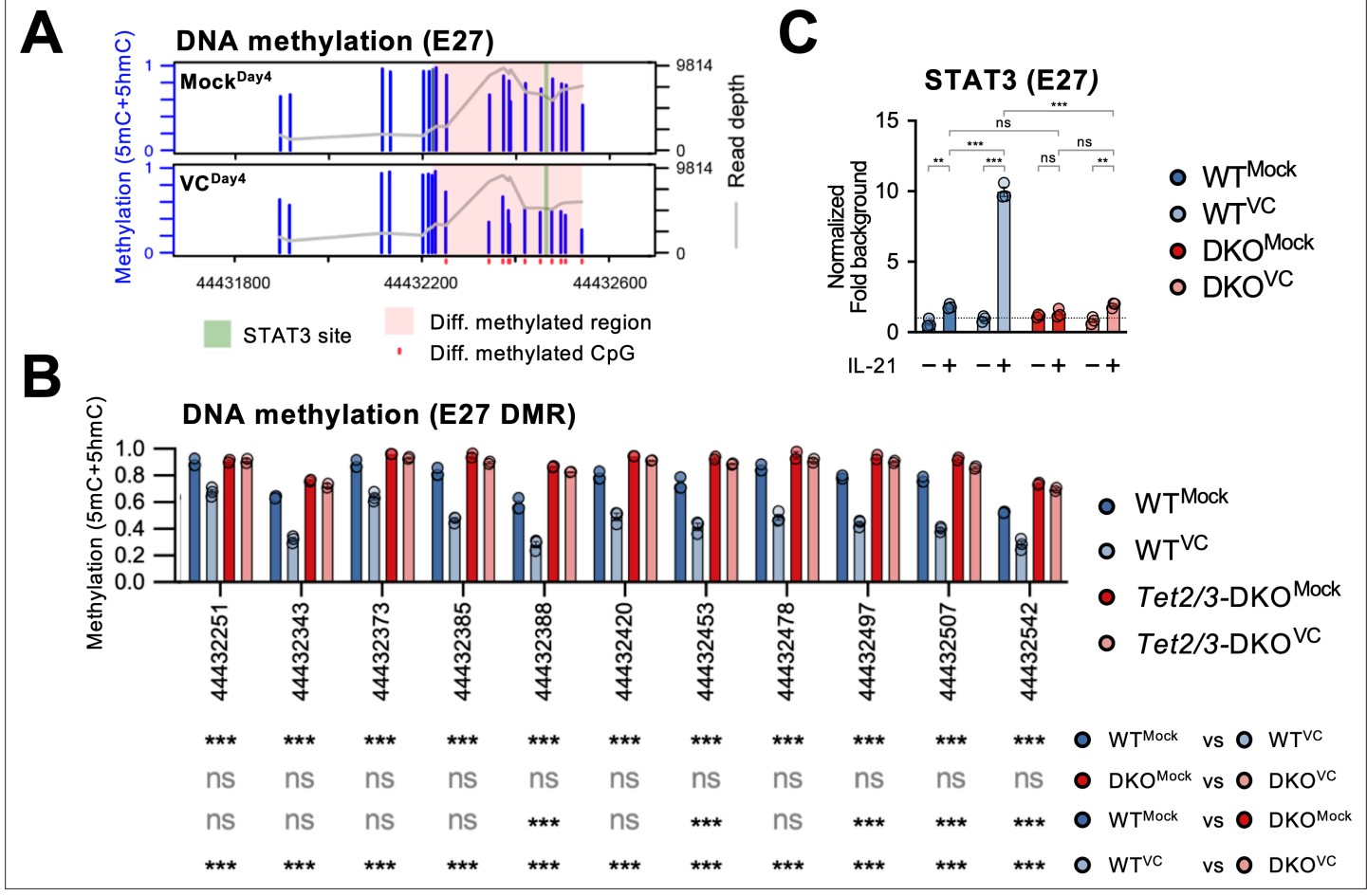

**Figure 9.** Vitamin C (VC)-facilitated DNA demethylation at E27 is required for efficient STAT3 recruitment. DNA methylation at E27 was analyzed using bisulfite amplicon sequencing with a Nanopore sequencer. Note that bisulfite sequencing is not able to distinguish 5mC and 5hmC, which usually represents a relatively minor fraction. (**A**) Differential DNA methylation region (DMR) at E27. The DNA methylation (5mC + 5hmC) levels were compared between control (Mock^Day4) and VC-treated B cells (VC^Day4). The average DNA methylation ratio at each CpG from three samples is plotted as blue bars (left Y-axis), and the coverage depth from sequencing is plotted as a gray line (right Y-axis). The differential DMR is highlighted with red and CpGs are indicated as red dots. Previously identified STAT3 binding site is highlighted in green. n = 3 for each group. (**B**) VC induces TET-mediated DNA demethylation at E27 DMR. DNA methylation ratios at E27 DMR from indicated groups are shown. Statistical significance was analyzed using a Bayesian hierarchical model with Wald test (*Park and Wu, 2016*). WT, n = 3; *Tet2/3*-DKO, n = 2. (**C**) TET2 and TET3 are required for the VC-facilitated STAT3 binding to E27. STAT3 binding to E27 was analyzed using ChIP-qPCR. Specific signal from E27 was normalized to the input and then with a background region ('Materials and methods'). Representative data from one of two experiments are shown (three technical replicates for each group). Statistical significance was calculated using two-way ANOVA, and the relevant comparisons are shown. ***p<0.001, **p<0.01. ns, not significant.

The online version of this article includes the following figure supplement(s) for figure 9:

**Figure supplement 1.** DNA methylation levels at the *Prdm1* locus.

**Figure supplement 2.** DNA sequence analysis of E27 differentially methylated region (DMR).

*2018*; *Douglas and Hemilä, 2005*; *Marik et al., 2017*; *Fisher et al., 2014*; *Wu et al., 2004*; *Hemilä, 2017*), relatively little is known about the mechanism about how VC deficiency may affect the immune response. Here, we provide an additional evidence that VC controls the differentiation of plasma cells that are essential for humoral immunity.

Using an in vitro two-step model of plasma cell differentiation, we performed a limited micronutrient analysis and identified VC as a strong potentiator of plasma cell differentiation in mouse and human B cells (*Figure 1C, D, F, and G*). The effect of VC on plasma cells is independent of its general antioxidant function (*Figure 1C–E*). We further identified that the main effect of VC occurred during the initial activation step (first step; *Figure 3A*), prior to IL-21 stimulation. Intriguingly, after 4 days of first-step culture, the percentage of plasma cells (*Figure 3—figure supplement 1*, *Figure 3B*),

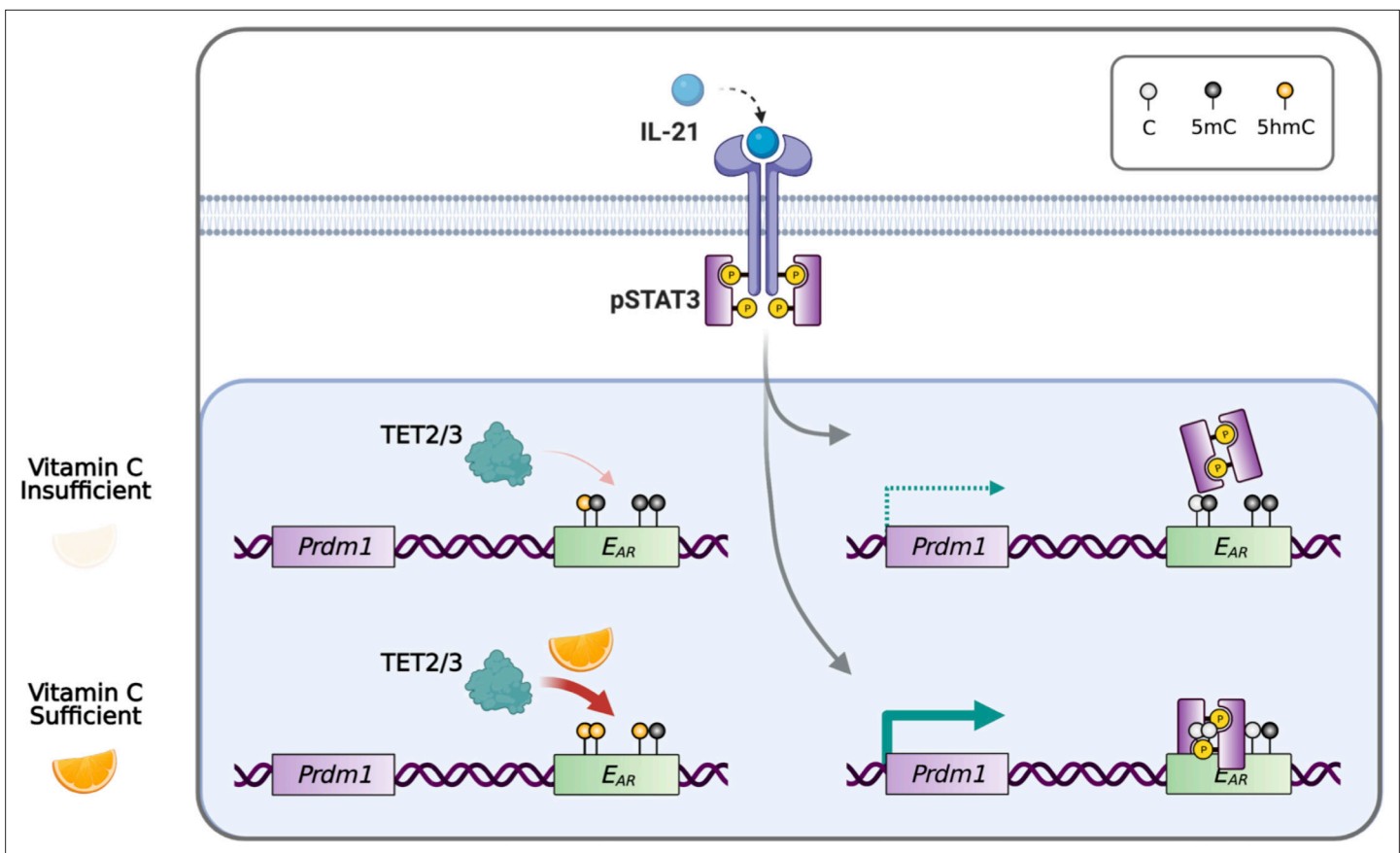

**Figure 10.** Model for vitamin C (VC)-enhanced plasma cell differentiation. During plasma cell differentiation, TET2 and/or TET3 is recruited to the enhancers of *Prdm1*. In the presence of sufficient VC, TET proteins have a higher enzymatic activity and efficiently oxidize 5mC into 5hmC at certain *cis* elements, which we termed 'ascorbate-responsive element' or E$_{AR}$. The oxidation and demethylation of 5mC at some E$_{AR}$ may increase the association of transcription factors (TFs), such as STAT3 to E27. However, when VC is limited, the decreased TET activity resulted in an inefficient oxidation of 5mC, which may preclude the binding of TF and/or the maintenance of the association between DNA and nucleosome (not depicted). Therefore, the activity of epigenetic enzyme may reflect the availability of micronutrients or metabolites and may influence gene expression and cell fate decisions.

capacity of proximal IL-21/STAT3 signaling (*Figure 4*), and transcriptomes (*Figure 5C*) were remarkably similar between control and VC-treated cells. As recent studies suggested that VC is a cofactor for α-ketoglutarate (αKG)-dependent epigenetic enzymes, we speculated that VC might support plasma cell differentiation by remodeling the epigenome via TET. Indeed, the positive effect of VC on plasma cells required TET2 and TET3 (*Figure 6*), and the analysis of the genome-wide 5hmC distribution revealed a significant difference between control and VC-treated cells (*Figure 7*). Our data demonstrated that VC facilitates plasma cell differentiation via TET.

The minimal difference in transcriptomes between control and VC groups on day 4 (*Figure 5C*) was surprising in light of the substantial effect of VC on plasma cell differentiation (*Figure 1*). The lack of difference in the transcriptome was likely not due to the sensitivity as we were able to detect a significant difference in RNA expression between activated and naïve B cells (*Figure 5A and B*). A previous study reported the coupled changes in DNA methylation and RNA expression at the later stage of plasma cell differentiation (*Barwick et al., 2016*). Given 5hmC modification precedes DNA demethylation, we would expect 5hmC modification to appear prior to transcriptional changes. Indeed, our results and another study on plasma cell differentiation (*Scharer et al., 2018*) support the notion that epigenome remodeling (e.g., 5hmC and chromatin accessibility) appears to precede transcriptional changes; and the changes in the epigenome may not necessarily result in immediate transcriptional responses.

Since *Prdm1*/BLIMP1 is the TF essential for plasma cell differentiation, we speculate that VC may affect the 5hmC modification at the *Prdm1* locus and render it permissive to STAT3-mediated

upregulation. Upon examining the 5hmC at the *Prdm1* locus, we identified at least 14 VC-induced DhmRs in the B cells cultured with VC (VC$^{Day4}$) that we termed 'ascorbate-responsive elements' (E$_{AR}$; *Figure 8B*), including the regions with gained 5hmC or DNA demethylation after VC treatment. One noticeable E$_{AR}$ is E58, an undescribed *cis*-element located at +58 kb (*Kwon et al., 2009*). While E58 contains a STAT3 motif and is permissive to STAT3 binding in T cells (*Kwon et al., 2009*), STAT3 does not associate with E58 in B cells in our system. Instead, we showed that STAT3 binds to the 3′ enhancer E27 and promoter. The binding was significantly increased in the presence of VC and correlated with DNA demethylation at E27, thus qualifying E27 as an E$_{AR}$ that undergoes VC-induced, TET-mediated DNA demethylation (*Figure 9*). Sequence analysis of E27 in mice and the corresponding region in humans showed that STAT3 and two of the CpG motifs are largely conserved (*Figure 9—figure supplement 2*). How the DNA methylation status of these neighboring CpGs affects STAT3 binding remains unclear. One possibility is that DNA methylation may decrease nucleosome accessibility (*Lio et al., 2016*; *Choy et al., 2010*), thus decreasing the STAT3 binding to the motif. The detailed mechanism remains to be addressed.

Our data is in accordance with a recent report showing that VC can promote plasma cell differentiation via TET enzymes (*Qi et al., 2020*). Similar to our results, Qi et al. showed that the effect of VC on plasma cells is dependent on TET, and TET-deficient affects VC-assisted DNA demethylation at the *Prdm1* intronic enhancers. While the previous study provided in vivo data, we have identified detailed molecular mechanisms of how VC enhances plasma cell differentiation. For instance, we dissected the two-step culture system and have shown that VC is critical during the initial B cell activation (first step). The effect of VC is unique and cannot be substituted by other antioxidants. Moreover, we demonstrated that VC has no immediate effect on the transcriptome and IL-21/STAT3 signaling after the first-step culture. Our data demonstrated that the main effect of VC is to remodel the epigenome. Furthermore, our genome-wide profiling of 5hmC revealed several E$_{AR}$ at the *Prdm1* locus, including a potential novel element E58 and the 3′ enhancer E27. Together, the studies here and Qi et al. have presented a clear picture of the importance of VC in regulating B cell differentiation.

DNA methylation is a dynamic process and undergoes substantial remodeling during differentiation. Studies have shown that differentiation from naïve B cells to plasma cells and memory B cells is accompanied by remodeling of the methylome, with more regions being hypomethylated that is consistent with the role of TET enzymes in cell differentiations (*Kulis et al., 2015*; *Oakes et al., 2016*; *Lai et al., 2013*). Therefore, these TET-dependent differentiations may be influenced by the levels of VC. We speculated that methylome remodeling may partially contribute to the increased effectiveness of memory B cells to differentiate into germinal center B cells and plasma cells. Interestingly, a previous report showed that human memory B cells, compared to naïve B cells, require a lower level of STAT3 to differentiate into plasma cells (*Deenick et al., 2013*). As DNA hypomethylation may enhance STAT3 binding, DNA demethylation may contribute to the lower requirement for STAT3 in memory B cells. In addition, given the lower methylation levels in memory compared to naïve B cells, we hypothesize that the differentiation of plasma cells from naïve B cells may be preferentially affected by VC deficiency.

VC has long been associated with immune function but the notion continues to be controversial. While VC supplementation does not prevent 'common cold,' a comprehensive meta-analysis of 29 clinical trials showed that VC supplementation is associated with shortened duration of symptoms (*Hemilä and Chalker, 2013*). However, how VC may enhance antiviral immunity remains unclear. Giving the importance of early IgM produced by antibody-secreting cells in controlling infection (*Nguyen et al., 2017*), it is possible that VC deficiency may delay the plasma cell differentiation to result in inferior control of the viruses at the early phase. Here, our in vitro findings provided a mechanistic explanation of how VC may affect humoral immune response. It is currently unclear how VC deficiency affects B cell function in vivo. Future experiments using genetic VC-deficient animal models (hamsters or *Gulo*-deficient mice) will provide critical insight into the physiological function of VC in immune response.

Increasing evidence has revealed the link between the metabolome and epigenome in the immune system (*Lio and Huang, 2020a*). For instance, VC stabilizes the in vitro-induced regulatory T cells by TET-mediated DNA demethylation at the intronic enhancer of the key TF *Foxp3* (*Sasidharan Nair et al., 2016*; *Someya et al., 2017*; *Yue et al., 2016*). Another example is αKG, an intermediate in the Kreb cycle and a substrate for many epigenetic enzymes. Studies have shown that αKG modulates

macrophage function by facilitating histone demethylation (*Liu et al., 2017*) and promoting IL-2-sensitive gene expression in T cells by TET-mediated DNA demethylation (*Yue et al., 2021*). Our results and others have shown that VC is a vital micronutrient required for efficient plasma cell differentiation (*Qi et al., 2020*). Based on these observations, we hypothesize that epigenetic enzymes may act as rheostats to control appropriate gene transcription by integrating the availability of micronutrients or metabolites. Future studies will be needed to address how micronutrients, including VC, contribute to immune response via epigenetic remodeling.

## Materials and methods

### Animal experiments

All animal experiments were approved by IACUCs at La Jolla Institute (AP00001025) and Ohio State University (2020A00000055). C57BL/6J mice of both sexes (6–12 weeks; Jax#000664) were purchased from Jackson Laboratory (Bar Harbor, ME). No significant difference was observed between cells isolated from male and female mice. *IgHCGG* mice were kindly provided by Dr. Gabriel Victora (The Rockefeller University). *Tet2*<sup>fl/fl</sup> *Tet3*<sup>fl/fl</sup> mice were provided by Dr. Anjana Rao (La Jolla Institute). *Tet2*<sup>fl/fl</sup> *Tet3*<sup>fl/fl</sup> mice crossed with *Rosa26*<sup>LSL-EYFP</sup> (Jax#006148) and *UBC*<sup>Cre-ERT2</sup> (Jax#008085) mice were described previously (*Lio et al., 2019*). To induce *Tet2/3*-deletion with *UBC*<sup>Cre-ERT2</sup>, mice were injected with 2 mg of tamoxifen in corn oil (MilliporeSigma, St. Louis, MO) for five consecutive days and rested for 2 days before the isolation of primary B cell.

### Cell culture

40LB cells were cultured in D10 media consisting of DMEM (high-glucose), 10% fetal bovine serum (FBS; Gemini Bio, West Sacramento, CA), and 2 mM GlutaMAX. Primary B cells were cultured in R10 media consisting of RPMI1640, 10% FBS (Gemini Bio), 2 mM GlutaMAX, 1× non-essential amino acid, 1 mM sodium pyruvate, 10 mM HEPES, 50 µg/mL gentamicin, and 55 µM 2-mercaptoethanol. B27 serum-free supplement (50×) was added as 1×. The components of B27 are listed in *Supplementary file 1d*. Most cell culture reagents above are from Gibco-Thermo Fisher (Waltham, MA). The authenticity of the 40LB cells was verified by their ability to stimulate B cell differentiation as in the original publication (*Nojima et al., 2011*). Cells were regularly tested for mycoplasma contamination and were negative.

### Induced germinal center (iGB or 40LB) culture

The iGB culture was performed according to previous publication with slight modifications (*Nishikimi et al., 1994*). Briefly, 40LB cells were irradiated (30 Gy) and 1 × 10<sup>5</sup> cells were plated for each well in a 24-well tissue-culture plate in D10 media. The following day, B cells were isolated from mouse spleens using EasySep Mouse B Cell Isolation Kit (STEMCELL Technologies, Vancouver, Canada) according to the manufacturer's instruction. For the first-step culture, B cells (10,000) were plated with the irradiated 40LB cells in R10 media with 1 ng/mL recombinant mouse interleukin-4 (rmIL-4; PeproTech, East Windsor, NJ). On day 4, B cells were resuspended and 10,000 cells were transferred to irradiated 40LB cells in R10 media with 10 ng/mL rmIL-21 (PeproTech) for the second-step culture. Cells were analyzed on day 7. For ascorbic acid treatment, L-ascorbic acid 2-phosphate (P-AA; MilliporeSigma), L-ascorbic acid (L-AA; MilliporeSigma) or EA (MilliporeSigma) were dissolved in water and sterilized with 0.22 um syringe filter. For ascorbic oxidase (AAO) treatment, 100 µM of P-AA (MilliporeSigma) or L-AA (MilliporeSigma) were treated with 0.1 U AAO (MilliporeSigma) for 1 hr before added into first-step co-culture. For VC treatment, P-AA (MilliporeSigma) was added at a final concentration of 100 µg/mL (310.5 µM). Unless stated otherwise, VC was added at the beginning of the first- and second-step co-culture. For the extended culture (11 days), B cells were collected from second-step-co-culture and 10,000 cells were transferred to irradiated 40LB cells in R10 media with 10 ng/mL rmIL-21 (PeproTech) for 4 days. Unless otherwise stated, VC used for tissue culture is the redox stable P-AA.

### LPS-stimulated B cells

Mouse B cells (500,000) were cultured for 4 days in 2 mL R10 media and stimulated with 50 µg/mL LPS (MilliporeSigma), LPS with 10 ng/mL rmIL-4 (PeproTech), or LPS with 10 U/mL recombinant mouse

IFN-γ (rmIFN-γ; PeproTech). B cells were analyzed by FACS on day 4. The culture supernatants were analyzed by ELISA for antibody secretion.

## Isolation and culture of primary human B cells

Human peripheral blood mononuclear cells (PBMCs) were isolated from healthy donors with IRB approval (2009H0314; to DJW). Briefly, 20–40 mL of whole blood was diluted with phosphate-buffered saline (PBS) and overlaid on Ficoll-Paque Plus (Cytiva, Marlborough, MA). After centrifugation, PBMCs were transferred from the interface and washed with PBS. Primary human B cells were isolated from PBMCs using the EasySep Human Naïve B Cell Isolation Kit (STEMCELL Technologies). The typical purity of naïve B cells is >95% (*Figure 1—figure supplement 5*). Naïve B cells were cultured with or without VC (100 µg/mL) in R10 media. To induce the differentiation of antibody-secreting cells, cells were cultured with recombinant human IL-21 (100 ng/mL; PeproTech) and either with:0.01% PANSORBIN (*S. aureus* Cowan I; Millipore; *Ettinger et al., 2005*); or ODN (2.5 µM; Invivogen, San Diego, CA) and anti-hCD40 (1 µg/mL; BioLegend). Cells were harvested on day 6 for flow cytometry analysis. Alternatively, cells were cultured with goat anti-human IgM/G F(ab')2 (2.6 µg/mL; Jackson ImmunoResearch), anti-hCD40 (100 ng/mL; BioLegend), CpG 1080 (1 µg/mL; TriLink), and rhIL-2 (50 U/mL; NIH) for 4 days. On day 4, B cells were washed and stimulated with rhIL-2 (50 U/mL), rhIL-10 (12.5 ng/mL; PeproTech), and rhIL-4 (5 ng/mL; PeproTech) for another 3 days (*Hipp et al., 2017*). Cells were analyzed by FACS on day 7.

## Flow cytometry

Cells were resuspended in FACS buffer (1% bovine serum albumin, 1 mM ethylenediaminetetraacetic acid [EDTA], 0.05% sodium azide) and incubated with Fc-receptor blocking antibody 2.4G2 (5 µg/mL; BioXCell, Lebanon, NH) for at least 10 min on ice. Fluorescence-conjugated antibodies and live-dead dye were added and incubated on ice for 25–30 min. Cells were washed at least two times with FACS buffer, fixed with 1% paraformaldehyde (Thermo Fisher), and analyzed by either FACS Canto-II, LSR-II, or Accuri C6 (all BD, Franklin Lakes, NJ). FACS Aria II (BD) and MA900 (Sony Biotechnology, San Jose, CA) were used for cell isolation. For intracellular transcription factor staining, cells were stained by using True-Nuclear transcription factor buffer set (BioLegend, San Diego, CA). Briefly, cells were fixed with 1× fix concentrate buffer (BioLegend) at room temperature (RT) for 45–60 min and washed twice with 1× permeabilization buffer. Cells were resuspended in 1× permeabilization buffer and stained with fluorescence-conjugated antibodies at RT for 30 min. Cells were washed twice with 1× permeabilization buffer and prior to FACS analysis. For detection of apoptosis, $1 \times 10^6$ cells were washed with PBS and resuspended in 1× binding buffer (10 mM HEPES pH 7.4, 140 mM NaCl, 2.5 mM CaCl$_2$). Biotin-conjugated Annexin V (STEMCELL Technologies) was added and incubated at RT for 10–15 min. Cells were washed and resuspended in 1× binding buffer. APC-Streptavidin (BioLegend) was added and incubated at RT for 10–15 min. Cells were washed, pelleted, and resuspended in 1× binding buffer. 7-Aminoactinomycin D (7-AAD; BD) was added to label the dead cells by incubating at RT for at least 5 min and analyzed by FACS immediately. FACS antibodies and reagents are listed in Key Resources Table.

## STAT3 phosphorylation assay

Cells from the first 40LB culture were washed once with complete media and stimulated with rmIL-21 (10 ng/mL final) at a concentration of $5 \times 10^5$ cells/mL at 37°C for 30 min. Cells were fixed immediately with paraformaldehyde (final 2%; Thermo Fisher) and incubated at 4°C for 30 min. Cells were washed twice with FACS buffer and the surface antigens were stained with fluorescence-conjugated antibodies at RT for 30 min. Cells were washed twice again with FACS buffer and fixed with ice-cold methanol (final 90%) and stored at –20°C overnight. The next day, cells were washed at least twice with FACS buffer to remove the fixative before incubating with the Fc-receptor blocking antibody 2.4G2 (BioXCell) and rat serum (STEMCELL Technologies), followed by the anti-phosphoSTAT3 (Tyr705) antibody (BioLegend) at RT for 30 min. Cells were washed twice with FACS buffer, fixed with 1% paraformaldehyde (Thermo Fisher), and analyzed by FACS.

## Cytosine 5-methylenesulfonate (CMS) dot blot

CMS dot blot was performed as previously described (*Webb and Villamor, 2007*). Briefly, genomic DNA was treated with MethylCode Bisulfite Conversion Kit (Thermo Fisher) to convert 5hmC into

CMS. DNA was diluted, denatured, neutralized, and immobilized on a nitrocellulose membrane with the Bio-Dot apparatus (Bio-Rad, Hercules, CA). CMS was detected using rabbit anti-CMS antisera (gift from Dr. Anjana Rao) followed by peroxidase-conjugated goat anti-rabbit IgG secondary antibody. Membrane was exposed using SuperSignal West Femto (Thermo Fisher) and X-ray films.

## 5hmC enrichment analysis
Genomic DNA was isolated from B cells either using Blood and Tissue or Flexigene kits (QIAGEN, Hilden, Germany). DNA was quantified with Qubit (Thermo Fisher) and sonicated to around 150–200 bp using Bioruptor Pico (Diagenode, Denville, NJ). 5hmC enrichment analysis was performed using the HMCP kit (Cambridge Epigenetix, Cambridge, UK) based on the addition of azido-glucose to 5hmC by T4-beta-glucosyltransferase following the manufacturer's instruction. The 5hmC-enriched and input libraries were sequenced using NovaSeq 6000 (Illumina, San Diego, CA) with 50 × 50 bp paired-end using SP flow cells (Nationwide Children Hospital, Columbus, OH).

## RNA sequencing (RNA-seq)
RNA was purified using RNeasy Kit (QIAGEN) and analyzed by TapeStation RNA tape (Agilent, Santa Clara, CA). Total RNA (RIN > 9.5) was used for mRNA isolation using the NEBNext Poly(A) mRNA Magnetic isolation module (NEB, Ipswich, MA). Libraries were constructed using NEBNext Ultra II Directional RNA Library Prep Kit (NEB) according to the manufacturer's protocol and barcoded using the NEBNext Multiplex (unique dual index) Oligos (NEB). Libraries were sequenced using NovaSeq 6000 (Illumina) with 50 × 50 bp paired-end using SP flow cells (Nationwide Children Hospital).

## Chromatin immunoprecipitation (ChIP)
To induce STAT3 binding, B cells cultured for 4 days with 40LB were stimulated with rmIL-21 (10 ng/mL) at $2 \times 10^6$ cells/mL at 37°C for 6 hr. ChIP was performed as previously described with modification (*Lio et al., 2019*). Briefly, cells were fixed with 1% formaldehyde (Thermo Fisher) in media at $1 \times 10^6$ cells/mL for 10 min at RT with constant nutation. Fixation was quenched with 125 mM glycine for 5 min on ice. After being washed twice with PBS, cell pellets were snap-frozen with liquid nitrogen and stored at −80°C until nuclei preparation. To extract chromatin, nuclei were isolated using lysis buffer (50 mM HEPES pH 7.5, 140 mM NaCl, 1 mM EDTA, 10% glycerol, 0.5% NP40, and 0.25% Triton X-100) and were washed once with wash buffer (10 mM Tris–HCl, pH 8.0, 200 mM NaCl, 1 mM EDTA, and 0.5 mM EGTA), followed by two washes with shearing buffer (10 mM Tris–HCl pH 8.0, 1 mM EDTA, and 0.1% SDS) without disturbing pellets. Pellets were then resuspended in shearing buffer and the chromatin was sheared using Bioruptor Pico (Diagenode) for seven cycles (30 s on, 30 s off) at 4°C. After precleared with ProteinA dynabeads (Thermo Fisher), 20 µg of chromatin was diluted to a final of 500 uL RIPA buffer (50 mM Tris–HCl, pH 8.0, 150 mM NaCl, 1 mM EDTA, 1% NP40, 0.1% SDS, and 0.5% sodium deoxycholate) and incubated with 1 µg rabbit monoclonal anti-phsopho-STAT3 (Tyr705) antibody (clone D3A7, Cell Signaling Technology) overnight at 4°C with constant rotation. To capture the antibody-bound chromatin, the samples were incubated with 30 µL of precleaned Protein A Dynabeads for 2 hr at 4°C. The beads were washed twice with RIPA buffer and once with TE buffer (10 mM Tris–HCl pH 8.0 and 1 mM EDTA). All washes were incubated for 5 min at 4°C with constant rotation. After the last wash, samples were eluted twice with 100 µL elution buffer (100 mM NaHCO$_3$, 1% SDS, and 0.5 mg/mL RNase A). To decrosslink, proteinase K (0.5 mg/mL; Ambion) and NaCl (200 mM) were added to the eluted samples, followed by the incubation at 65°C overnight. The samples were purified using DNA Clean & Concentrator-5 (Zymo Research). ChIP-seq libraries were prepared using NEBNext Ultra II following the instruction and sequenced as described above for RNA-seq. For ChIP-qPCR, eluted DNA was diluted and used as templates for qPCR using 2X SYBR Select Master Mix for CFX (Applied Biosystems). The assay was performed using the CFX96 real-time PCR detection system (Bio-Rad). The enrichment of ChIP DNA was normalized first to the amount of input followed by the nonspecific background at a negative control locus (*Cd4* promoter). ChIP-qPCR primers are listed in Key Resources Table.

## ELISA
Culture supernatants were collected on days 4 and 7 of 40LB culture and stored at –20°C. Capture antibodies for specific isotypes (goat anti-IgM, -IgG, and IgE; SouthernBiotech, Birmingham, AL)

were coated on high-binding 96-well plates (Corning, Tweksbury, MA) at 1 µg/mL overnight. After blocking with 1% bovine serum albumin in PBS, standards (IgM and IgG from SouthernBiotech; IgE from BioLegend) and samples were applied to the coated plate for 1 hr at RT. After washed with PBS-T (PBS with 0.05% Tween-20), 160 ng/mL of HRP-conjugated donkey anti-mouse IgG (H+L) detection antibody (Jackson ImmunoResearch, West Grove, PA) was added to the plates and incubated for 1 hr at RT. The peroxidase substrate tetramethylbenzidine (TMB; Thermo Fisher) was added and the reactions were stopped by adding 50 uL of 1 M sulfuric acid. The absorbance (OD) of the plates was read using SpectraMax (Molecular Devices, San Jose, CA) at 450 nm (using OD at 540 nm as background). Antibody concentrations were extrapolated from the standards.

## Oxidative stress analysis

The ROS levels were detected using CellROX deep red flow cytometry assay kit (Thermo Fisher) according to the manufacturer's protocol. Briefly, 50,000 cells cultured for 4 days with 40LB were added to a 24-well plate. Cells might be treated with 250 uM NAC for 1 hr at 37°C. TBHP (200 µM), an inducer of ROS, might be added to the cells for 30 min 37°C. To monitor the oxidative stress, 2 uL of CellROX Deep Red reagent was added to the cells for 15 min followed by 1 uL of SYTOX Blue Dead Cell stain solution for another 15 min. The cells were analyzed immediately by flow cytometry.

## Cell sorting

Cre$^{ERT2}$ Tet2$^{fl/fl}$ Tet3$^{fl/fl}$ Rosa26$^{LSL-EYFP}$ mice and Cre$^{ERT2}$ Rosa26$^{LSL-EYFP}$ (control) were injected intraperitoneally with tamoxifen (Sigma; 2 mg per mouse in corn oil) for five consecutive days and rested for 2 days before the isolation of primary B cell. CD19$^+$ YFP$^+$ live splenic B cells were sorted using MA900 with 100 µm flow chips (Sony Biotechnology).

## Nanopore bisulfite DNA sequencing

Purified genomic DNA was treated with bisulfite following the manufacturer's protocol (EpiJET Bisulfite Conversion Kit, Thermo Fisher) and used as templates for PCR (PyroMark, QIAGEN) using the primers listed in Key Resources Table. For nanopore sequencing, the PCR products were pooled, labeled with the native barcodes (EXP-NBD104 Native Barcoding Expansion, Oxford Nanopore), and sequenced using the Flongle Flow Cell (R9.4.1, Oxford Nanopore) with a MinION sequencer. Reagents are listed in Key Resources Table.

## Bioinformatics analyses

### ChIP and 5hmC analysis

Paired-end reads were mapped to the mouse genome mm10 GRCm38 (December 2011) from UCSC using Bowtie 2 (v2.4.2; *Langmead and Salzberg, 2012*), and reads that failed the alignment and duplicates were removed using Samtools (v1.10). Transcripts mapping to autosomal and sex chromosomes were kept. BigWig files were generated using bamCoverage (--normalizeUsing RPKM) using deepTools (v3.5.1; *Ramírez et al., 2016*). For 5hmC, peaks were called using MACS2 (*Zhang et al., 2008*) with control samples (callpeak -g 1.87e9 -q 0.01 `--keep-dup` all `--nomodel` –broad). Reads were compared to the blacklisted regions for mm10 (ENCODE 2016). Count matrix was generated using DiffBind (v2.0.2; *Ross-Innes et al., 2012*) dba count with normalization DBA_SCORE_TMM_ MINUS_EFFECTIVE. Differentially enriched 5hmC regions were determined using edgeR; regions were selected by an adjusted p-value<0.05. Heatmaps were generated using deepTools.

### RNA-seq analysis

Paired-end reads were mapped to the mouse genome using GRCm38 genome sequence, primary assembly (ENCODE 2016) using STAR (v 2.7.0). Gene count matrix was generated with STAR using Comprehensive gene annotation CHR Regions. Unmapped regions were filtered. Differential gene expression was calculated using limma (v3.12) using voom transformation. Genes were selected by an adjusted p-value (false discovery rate [FDR]) <0.05 and a log(2) fold enrichment ≥1.

### Published STAT3 ChIP-seq

The BigWig files for STAT3 ChIP-seq were downloaded from Cistrome (CistromeDB # 4577 and 4580). The data were from CD4 T cells stimulated with IL-21 (100 ng/mL) for 1 hr (*Kwon et al., 2009*).

## Nanopore base calling and methylation analysis

FAST5 files were converted to FASTQ with Guppy v6.1.2 (--config dna_r9.4.1_450bps_hac.cfg) followed by demultiplexing. Porechop (v0.2.4; *Wick et al., 2017*) and NanoFilt (v2.8.0; *De Coster et al., 2018*) were used to remove adapters and reads with lower average quality (q12), respectively. DNA methylation analysis was performed by the nanoEM pipeline (*Sakamoto et al., 2021*). Minimap2 (v2.24; *Li, 2018*) was used for mapping the in silico-converted reads to the converted reference genome mm10 with the preset for Oxford Nanopore genomic reads (*map-ont*). The corresponding unconverted reads were isolated from the original FASTQ files with best_align.py (*Sakamoto et al., 2021*). Reads were then used to call methylation with Samtools (v1.15.1) *mpileup* (*Li et al., 2009*). CpGs with fewer than 100× coverages were removed from downstream analysis. Differentially methylation regions and CpGs were analyzed using DSS (Dispersion Shrinkage for Sequencing data, v2.44.0) package (*Park and Wu, 2016*) in R (v4.0.2), with *smoothing=OFF* for *DMLtest*. The threshold for calling differentially methylated CpGs (*callDML*) was p<0.001.

## Acknowledgements

Drs. Anjana Rao, Patrick Hogan (LJI), Andrew J McKnight, Rachel Soloff, Ann-Laure Perraud (Kyowa Kirin, Inc, formerly Kyowa Kirin Pharmaceutical Research; KKR) for the support and advice; Roberta Nowak and Samantha Blake for assistance (LJI); Cheryl Kim and Denise Hinz at LJI for cell sorting; the Institute for Genomic Medicine at Nationwide Children's Hospital for NGS-sequencing; Drs. Eugene Oltz and Ken Oestreich (OSU) for advice and critical reading; Dr. Hazem Ghoneim for assisting with experiments; Drs. Emily Hemann, Gang Xin, and Adriana Forero (OSU) for providing critical reagents during the peak of the pandemic; and all the Lio lab members for their contribution and critical reading of the manuscript. 40LB cells were a gift from Dr. Daisuke Kitamura (Tokyo University of Science). IgHCGG mice were provided by Dr. Gabriel Victora (The Rockefeller University). This research was funded by LJI/KKR independent investigator fund; NIH National Cancer Institute K22 (K22CA241290); startup funds from the Department of Microbial Infection and Immunity and from the Pelotonia Institute of Immuno-oncology at the Ohio State University (all to C-WJL).

## Additional information

### Funding

| Funder | Grant reference number | Author |
| --- | --- | --- |
| National Cancer Institute | K22CA241290 | Chan-Wang Jerry Lio |
| La Jolla Institute for Immunology | | Chan-Wang Jerry Lio |
| Kyowa Kirin, Inc | | Chan-Wang Jerry Lio |

The funders had no role in study design, data collection and interpretation, or the decision to submit the work for publication.

### Author contributions

Heng-Yi Chen, Data curation, Formal analysis, Investigation, Visualization, Writing – original draft, Writing – review and editing, Performed most experiments; Ana Almonte-Loya, Data curation, Software, Formal analysis, Investigation, Visualization, Writing – original draft, Writing – review and editing, Performed bioinformatics analyses; Fang-Yun Lay, Data curation, Formal analysis, Investigation, Performed human cell experiments; Michael Hsu, Data curation, Formal analysis, Writing – review and editing, Performed Nanopore bisulfite sequencing and related bioinformatics analyses; Eric Johnson, Jieyun Yin, Investigation; Edahí González-Avalos, Software, Advice on bioinformatics analyses; Richard S Bruno, Validation, Writing – review and editing; Qin Ma, Software, Advice on bioinformatics analyses; Hazem E Ghoneim, Resources, Validation; Daniel J Wozniak, Resources; Fiona E Harrison, Resources, Writing – review and editing; Chan-Wang Jerry Lio, Conceptualization, Resources, Data curation, Formal analysis, Supervision, Funding acquisition, Investigation, Visualization, Writing – original draft, Project administration, Writing – review and editing

## Author ORCIDs

Heng-Yi Chen ⬥ http://orcid.org/0000-0001-9283-6144
Michael Hsu ⬥ http://orcid.org/0000-0003-3852-1461
Chan-Wang Jerry Lio ⬥ http://orcid.org/0000-0003-3876-6741

## Ethics

Human subjects: Primary human peripheral blood was obtained from healthy adult donors according to the protocol approved by The Ohio State University Biomedical Sciences Institutional Review Board (2009H0314). Informed written consent was obtained from all donors.

This study was performed in accordance with the recommendations in the Guide for the Care and Use of Laboratory Animals of the National Institutes of Health. All of the animals were handled according to approved institutional animal care and use committee (IACUC) protocols of the La Jolla Institute (AP00001025) and Ohio State University (2020A00000055).

## Decision letter and Author response

Decision letter https://doi.org/10.7554/eLife.73754.sa1
Author response https://doi.org/10.7554/eLife.73754.sa2

## Additional files

### Supplementary files

• Transparent reporting form

• Supplementary file 1. Supplementary tables. (a) Differentially expressed genes (DEGs) between naïve and day 4 mock B cells. (b) DEGs between naïve and day 4 vitamin C (VC) B cells. (c) DEGs between day 4 mock and day 4 VC B cells. (d) Components of B27 serum-free supplement.

### Data availability

The data have been deposited to National Center for Biotechnology Information Gene Expression Omnibus (GEO GSE183681).

The following dataset was generated:

| Author(s) | Year | Dataset title | Dataset URL | Database and Identifier |
|---|---|---|---|---|
| Chen et al | 2021 | Vitamin C Potentiates Plasma Cell Differentiation via TET-mediated DNA modification | https://www.ncbi.nlm.nih.gov/geo/query/acc.cgi?acc=GSE183681 | NCBI Gene Expression Omnibus, GSE183681 |

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

# Appendix 1

## Appendix 1—key resources table

| Reagent type (species) or resource | Designation | Source or reference | Identifiers | Additional information |
|---|---|---|---|---|
| Antibody | PE/Cyanine7 anti-mouse CD138 antibody (rat monoclonal) | BioLegend | Cat# 142514; RRID:AB_2562198 | FACS (1:500) |
| Antibody | Brilliant Violet 570 anti-mouse CD19 antibody (rat monoclonal) | BioLegend | Cat# 115535; RRID:AB_10933260 | FACS (1:200) |
| Antibody | PE/Cyanine7 anti-human CD38 antibody (HIT2) (mouse monoclonal) | BioLegend | Cat# 303516; RRID:AB_2072782 | FACS (1.5 uL per test) |
| Antibody | Alexa Fluor 488 anti-human IgD antibody (mouse monoclonal) | BioLegend | Cat# 348216; RRID:AB_11150595 | FACS (1.5 uL per test) |
| Antibody | APC anti-human CD19 antibody (HIB19) (mouse monoclonal) | BioLegend | Cat# 302212; RRID:AB_314242 | FACS (1.5 uL per test) |
| Antibody | Alexa Fluor 647 anti-mouse Blimp-1 antibody (rat monoclonal) | BioLegend | Cat# 150004; RRID:AB_2565618 | FACS (1:100) |
| Antibody | Alexa Fluor 488 anti-Pax-5 antibody (rat monoclonal) | BioLegend | Cat# 649705; RRID:AB_2562426 | FACS (1:100) |
| Antibody | Pacific Blue anti-IRF4 antibody (rat monoclonal) | BioLegend | Cat# 646418; RRID:AB_2814497 | FACS (1:100) |
| Antibody | PE anti-mouse CD138 (Syndecan-1) antibody (rat monoclonal) | BioLegend | Cat# 142504; RRID:AB_10916119 | FACS (1:500) |
| Antibody | PE anti-mouse IL-21R antibody (rat monoclonal) | BioLegend | Cat# 131905; RRID:AB_1279431 | FACS (1:200) |
| Antibody | Alexa Fluor 647 anti-STAT3 phospho (Tyr705) antibody (mouse monoclonal) | BioLegend | Cat# 651007; RRID:AB_2572085 | FACS (1:20) |
| Antibody | PE anti-mouse Blimp-1 antibody (rat monoclonal) | BioLegend | Cat# 150005; RRID:AB_2565991 | FACS (1:100) |
| Antibody | Alexa Fluor 647 anti-Pax-5 antibody (rat monoclonal) | BioLegend | Cat# 649703; RRID:AB_2562424 | FACS (1:100) |
| Antibody | Anti-CMS antisera (rabbit polyclonal) | Dr. Anjana Rao | PMID:23018193 | Dot blot (1:3000) |
| Antibody | Goat anti-mouse IgM-UNLB 1 mg (goat polyclonal) | SouthernBiotech | Cat# 1021-01; RRID:AB_2687524 | ELISA (1:1000) |
| Antibody | Goat anti-mouse IgG Fc-UNLB 1 mg (goat polyclonal) | SouthernBiotech | Cat# 1033-01; RRID:AB_2794330 | ELISA (1:1000) |
| Antibody | AffiniPure donkey anti-mouse IgG (H+L) HRP (donkey polyclonal) | Jackson ImmunoResearch | Cat# 715-035-150; RRID:AB_2340770 | ELISA (1:5000) |
| Antibody | Goat anti-mouse IgG Fc-biotin (goat polyclonal) | SouthernBiotech | Cat# 1033-08; RRID:AB_2794333 | ELISA (1:5000) |
| Antibody | Phospho-Stat3 (Tyr705) (D3A7) XP (rabbit monoclonal) | Cell Signaling Technology | Cat# 9145; RRID:AB_2491009 | ChIP (10 μL) |
| Antibody | Ultra-LEAF purified anti-human CD40 (mouse monoclonal) | BioLegend | Cat# 334350; RRID:AB_2810512 | 1 or 100 ng/mL |
| Antibody | PE/Cyanine7 anti-mouse CD138 antibody (rat monoclonal) | BioLegend | Cat# 142514; RRID:AB_2562198 | FACS (1:500) |
| Antibody | AffiniPure F(ab')₂ fragment goat anti-human IgG +IgM (H+L) (goat polyclonal) | Jackson ImmunoResearch | Cat# 109-006-127; RRID:AB_2337552 | 2.6 μg/mL |
| Antibody | Anti-mouse CD16/CD32 antibody 2.4G2 (rat monoclonal) | BioXCell | Cat# BE0307; RRID:AB_2736987 | FACS (1:100) |
| Cell line (*Mus musculus*) | 40LB | *Nojima et al., 2011* | PMID:21897376 | |
| Chemical compound or drug | Gibco DMEM | Thermo Fisher | Cat# 11995-065 | |
| Chemical compound or drug | Fetal bovine serum (FBS) | Gemini Bio | Cat# 100-106 | |

*Appendix 1 Continued on next page*

*Appendix 1 Continued*

| Reagent type (species) or resource | Designation | Source or reference | Identifiers | Additional information |
|---|---|---|---|---|
| Chemical compound or drug | GlutaMAX | Thermo Fisher | Cat# 35050061 | |
| Chemical compound or drug | RPMI 1640 | Thermo Fisher | Cat# 61870127 | |
| Chemical compound or drug | MEM non-essential amino acids solution (100×) | Thermo Fisher | Cat# 11140050 | |
| Chemical compound or drug | Sodium pyruvate (100 mM) | Thermo Fisher | Cat# 25-000CI | |
| Chemical compound or drug | Gentamicin (50 mg/mL) | Thermo Fisher | Cat# 15750060 | |
| Chemical compound or drug | 2-Mercaptoethanol | MilliporeSigma | Cat# M3701 | |
| Chemical compound or drug | B-27 supplement (50×), serum free | Thermo Fisher | Cat# 17504044 | |
| Chemical compound or drug | B-27 supplement (50×), minus antioxidants | Thermo Fisher | Cat# 10889038 | |
| Chemical compound or drug | L-ascorbic acid 2-phosphate (P-AA; VC) | MilliporeSigma | Cat# 49572 | |
| Chemical compound or drug | L-ascorbic acid (L-AA) | MilliporeSigma | Cat# A92902 | |
| Chemical compound or drug | Erythorbic acid (EA) | MilliporeSigma | Cat# 856061 | |
| Chemical compound or drug | Lipopolysaccharides from *Escherichia coli* O55:B5 purified by phenol extraction | MilliporeSigma | Cat# L2880 | |
| Chemical compound or drug | Phosphate-based saline (PBS) | Thermo Fisher | Cat# 10010-023 | |
| Chemical compound or drug | Ficoll-Paque Plus | Cytiva | Cat# 17144002 | |
| Chemical compound or drug | PANSORBIN (*S. aureus* Cowan I) | MilliporeSigma | Cat# 507862 | 0.01% |
| Chemical compound or drug | ODN 2006 (ODN 7909) | Invivogen | Cat# tlrl-2006 | 2.5 µM |
| Chemical compound or drug | HEPES (1 M) | Thermo Fisher | Cat# 15630080 | 10 mM |
| Chemical compound or drug | 7-Aminoactinomycin D (7-AAD) | BD | Cat# 559925 | |
| Chemical compound or drug | Paraformaldehyde | Thermo Fisher | Cat# J61899.AK | |
| Chemical compound or drug | Rat serum | STEMCELL Technologies | Cat# 13551 | |
| Chemical compound or drug | 16% formaldehyde (w/v), methanol-free | Thermo Fisher | Cat# 28906 | 1% |
| Chemical compound or drug | Glycine | Fisher | Cat# 50-751-6880 | 125 mM |
| Chemical compound or drug | EDTA | Fisher | Cat# BP120-500 | |
| Chemical compound or drug | Glycerol | Fisher | Cat# BP229-1 | |
| Chemical compound or drug | Triton X-100 | MilliporeSigma | Cat# T8787 | |
| Chemical compound or drug | Tris–HCl | MilliporeSigma | Cat# T3253 | |
| Chemical compound or drug | Sodium dodecyl sulfate (SDS) | Fisher | Cat# BP166-500 | |

*Appendix 1 Continued on next page*

*Appendix 1 Continued*

| Reagent type (species) or resource | Designation | Source or reference | Identifiers | Additional information |
|---|---|---|---|---|
| Chemical compound or drug | Protein A dynabeads | Thermo Fisher | Cat# 10013D | |
| Chemical compound or drug | Sodium bicarbonate | MilliporeSigma | Cat# S5761 | |
| Chemical compound or drug | Tamoxifen | MilliporeSigma | Cat# 10540-29-1 | 2 mg per mouse in corn oil |
| Commercial assay or kit | eBioscience Fixable Viability Dye eFluor 780 | eBioscience | Cat# 65-0865-14 | FACS (1:1000) |
| Commercial assay or kit | CellROX Deep Red Flow Cytometry Assay Kit | Thermo Scientific | Cat# C10491 | |
| Commercial assay or kit | Biotin Annexin V | Stem Cell Technology | Cat# 17899C | FACS (1:200) |
| Commercial assay or kit | APC Streptavidin | BioLegend | Cat# 405207 | FACS (1:200) |
| Commercial assay or kit | Mouse IgE ELISA MAX Capture Antibody | BioLegend | Cat# 79122 | ELISA (1:200) |
| Commercial assay or kit | Mouse IgE ELISA MAX Detection Antibody | BioLegend | Cat# 79123 | ELISA (1:200) |
| Commercial assay or kit | Avdin-HRP | BioLegend | Cat# 79004 | ELISA (1:5000) |
| Commercial assay or kit | Mouse IgE Standard | BioLegend | Cat# 401801 | ELISA (1:200) |
| Commercial assay or kit | EpiJET Bisulfite Conversion Kit | Thermo Scientific | Cat# K1461 | |
| Commercial assay or kit | PyroMark PCR Kit | QIAGEN | Cat# 978703 | |
| Commercial assay or kit | Ligation Sequencing Kit | Nanopore | Cat# SQK-LSK110 | |
| Commercial assay or kit | Native Barcoding Expansion 1–12 (PCR-free) | Nanopore | Cat# EXP-NBD104 | |
| Commercial assay or kit | Flongle Flow Cell (R9.4.1) | Nanopore | Cat# FLO-FLG001 | |
| Commercial assay or kit | EasySep Mouse B Cell Isolation Kit | STEMCELL Technologies | Cat# 19854A | |
| Commercial assay or kit | EasySep Human Naïve B Cell Isolation Kit | STEMCELL Technologies | Cat# 17254 | |
| Commercial assay or kit | MethylCode Bisulfite Conversion Kit | Thermo Fisher | Cat# MECOV50 | |
| Commercial assay or kit | RNeasy Kit | QIAGEN | Cat# 74004 | |
| Commercial assay or kit | D1000 Screen Tape | Agilent | Cat# 5067-5582 | |
| Commercial assay or kit | D1000 Sample Buffer | Agilent | Cat# 5067-5583 | |
| Commercial assay or kit | D1000 Ladder | Agilent | Cat# 5067-5586 | |
| Commercial assay or kit | RNA Screen Tape | Agilent | Cat# 5067-5576 | |
| Commercial assay or kit | RNA Screentape Sample Buffer | Agilent | Cat# 5067-5577 | |
| Commercial assay or kit | NEBNext Poly(A) mRNA Magnetic Isolation Module | NEB | Cat# E7490 | |
| Commercial assay or kit | NEBNext Ultra II Directional RNA Library Prep Kit | NEB | Cat# E7760L | |
| Commercial assay or kit | NEBNext Multiplex Oligos for Illumina (Dual Index Primers Set) | NEB | Cat# E7600S, E7780S | |

*Appendix 1 Continued on next page*

*Appendix 1 Continued*

| Reagent type (species) or resource | Designation | Source or reference | Identifiers | Additional information |
|---|---|---|---|---|
| Commercial assay or kit | DNA Clean & Concentrator-5 | Zymo Research | Cat# NC9552153 | |
| Commercial assay or kit | 2X SYBR Select Master Mix for CFX | Applied Biosystems | Cat# 4472942 | |
| Gene (*M. musculus*) | *Prdm1* | MGI:99655; NCBI Gene: 12142 | | |
| Genetic reagent (*M. musculus*) | IgHCGG | Dr. Gabriel Victora | PMID:30181412 | |
| Genetic reagent (*M. musculus*) | *Tet2*^fl/fl^*Tet3*^fl/fl^ | Dr. Anjana Rao | N/A | |
| Genetic reagent (*M. musculus*) | *Rosa26*^LSL-EYFP^ | The Jackson Laboratory | Cat# 006148 | |
| Genetic reagent (*M. musculus*) | Ubc-Cre^ERT2^ | The Jackson Laboratory | Cat# 008085 | |
| Genetic reagent (*M. musculus*) | C57BL/6J | The Jackson Laboratory | Cat# 000664 | |
| Peptide, recombinant protein | Recombinant human interleukin-21 (rhIL-21) | PeproTech | Cat# 200-21 | 100 ng/mL |
| Peptide, recombinant protein | Mouse IgM Standard | SouthernBiotech | Cat# 5300-01B | ELISA (1:100) |
| Peptide, recombinant protein | Mouse IgG1 Standard | SouthernBiotech | Cat# 5300-01B | ELISA (1:10000) |
| Peptide, recombinant protein | NEBNext Ultra II End Repair/dA-Tailing | NEB | Cat# E7546L | |
| Peptide, recombinant protein | NEB Blunt/TA Ligase Master Mix | NEB | Cat# M0367 | |
| Peptide, recombinant protein | Recombinant murine interleukin-4 (rmIL-4) | PeproTech | Cat# 214-14 | 40LB-B: 1 ng/mL; LPS: 10 ng/mL |
| Peptide, recombinant protein | Recombinant murine interleukin-21 (rmIL-21) | PeproTech | Cat# 210-21 | 40LB-B: 10 ng/mL |
| Peptide, recombinant protein | Ascorbate oxidase, *Cucurbita* sp. (1000 U) | MilliporeSigma | Cat# 189724 | 0.1 U |
| Peptide, recombinant protein | Recombinant murine IFN-γ (rmIFN-γ) | PeproTech | Cat# 315-05 | 10 U/mL |
| Peptide, recombinant protein | Recombinant human interleukin-2 (rhIL-2) | NIH | N/A | 50 U/mL |
| Peptide, recombinant protein | Recombinant human interleukin-10 (rhIL-10) | PeproTech | Cat# 200-10 | 12.5 ng/mL |
| Peptide, recombinant protein | Recombinant human interleukin-4 (rhIL-4) | PeproTech | Cat# 200-04 | 5 ng/mL |
| Peptide, recombinant protein | RNase A | Thermo Fisher | Cat# R1253 | |
| Peptide, recombinant protein | Proteinase K | QIAGEN | Cat# 19131 | 0.5 mg/mL |
| Peptide, recombinant protein | Trypsin-EDTA (0.05%), phenol red | Thermo Fisher | Cat# 25300120 | |
| Sequence-based reagent | CpG 1080 (phosphorothioate backbone) | TriLink | N/A | 1 µg/mL; TGACTGTGAACGTTCGAGATGA |
| Sequence-based reagent | Prdm1-Pro-BS-F1 | IDT | Bisulfite PCR primers | AGAGAAGATTTAATATTTGAGATAAGTT |
| Sequence-based reagent | Prdm1-Pro-BS-R1 | IDT | Bisulfite PCR primers | CAATCCTTATTAAAATCCATTTACAAAC |
| Sequence-based reagent | Prdm1-E27-BS-F1 | IDT | Bisulfite PCR primers | GTGTGTATTTGAGTGTTTTTTTTAATAT |
| Sequence-based reagent | Prdm1-E27-BS-R1 | IDT | Bisulfite PCR primers | CTAACCTCAAATCCTATCTATATTAACA |

*Appendix 1 Continued on next page*

*Appendix 1 Continued*

| Reagent type (species) or resource | Designation | Source or reference | Identifiers | Additional information |
|---|---|---|---|---|
| Sequence-based reagent | Prdm1-E27-BS-F2 | IDT | Bisulfite PCR primers | AATATAGATAGGATTTGAGGTTAGGTTA |
| Sequence-based reagent | Prdm1-E27-BS-R2 | IDT | Bisulfite PCR primers | TATAACAAAAAAACTAACCTAAACAACC |
| Sequence-based reagent | Prdm1-E27-BS-F3 | IDT | Bisulfite PCR primers | GTAAAATGGTTTATATTATTTGTGTTGG |
| Sequence-based reagent | Prdm1-E27-BS-R3 | IDT | Bisulfite PCR primers | AAAAAAAATTAAAACCAAAACAAAAACT |
| Sequence-based reagent | Prdm1-E58-BS-F1 | IDT | Bisulfite PCR primers | GTAGGTTTTTTTGTTTGTTTAGTATTA |
| Sequence-based reagent | Prdm1-E58-BS-R1 | IDT | Bisulfite PCR primers | CCTTAATCACTAACTCAATATAAAACAA |
| Sequence-based reagent | Prdm1-E58-BS-F2 | IDT | Bisulfite PCR primers | TTTATATTGAGTTAGTGATTAAGGTGAA |
| Sequence-based reagent | Prdm1-E58-BS-R2 | IDT | Bisulfite PCR primers | CCTTAAAAACCTTATATAAACCCATAAC |
| Sequence-based reagent | Prdm1-E58-BS-F3 | IDT | Bisulfite PCR primers | ATAAGAGATAGTTTATGGTTTTAAGGAG |
| Sequence-based reagent | Prdm1-E58-BS-R3 | IDT | Bisulfite PCR primers | AAACTAAACTATCACTATCTAACTAACA |
| Sequence-based reagent | Prdm1-E58-BS-F4 | IDT | Bisulfite PCR primers | TTTTGTGTGATTTTTTAGATAAGTAAGT |
| Sequence-based reagent | Prdm1-E58-BS-R4 | IDT | Bisulfite PCR primers | ACTCTACCTATAATACTAAACAAACAAA |
| Sequence-based reagent | Cd4-Ch-F | IDT | ChIP-qPCR primers | CCCATAGGGAAACAGCAAGA |
| Sequence-based reagent | Cd4-Ch-R | IDT | ChIP-qPCR primers | CCCACTCAATCTCCAGCAAT |
| Sequence-based reagent | Prdm1-E27-Ch-F | IDT | ChIP-qPCR primers | CAGTGCAGCAGTGGAGGTTA |
| Sequence-based reagent | Prdm1-E27-Ch-R | IDT | ChIP-qPCR primers | AACCGTTGAAAGACGGTGAC |
| Software, algorithm | FlowJo V10 | TreeStar | RRID:SCR_008520 | Flow data processing and analysis |
| Software, algorithm | GraphPad Prism V8 | GraphPad | RRID:SCR_002798 | Graphs and statistical analysis |

