## [Editor Report]

This article describes the role of vitamin C in promoting plasma cell differentiation by remodeling the epigenome via TET family proteins. Activated Tet2/Tet3 actively demethylate selected genomic regions, including the genetic locus encoding Blimp-1, the master transcription factor of plasma cell differentiation. This article will be of interest to scientists in molecular immunologists, particularly those involved in epigenetic mechanisms of B cell differentiation to plasma cells.

---

## [Decision Letter]

**Decision letter after peer review:**

Thank you for submitting your article "Epigenetic Remodeling by Vitamin C Potentiates the Differentiation of Mouse and Human Plasma Cells" for consideration by *eLife*. Your article has been reviewed by 3 peer reviewers, and the evaluation has been overseen by a Reviewing Editor and Betty Diamond as the Senior Editor. The following individuals involved in review of your submission have agreed to reveal their identity: Paolo Casali (Reviewer #2); Frances Lee (Reviewer #3).

Essential revisions:

For revision, we would like the authors to focus on the following:

(1) Provide ChIP-seq data on Stat3 binding in the Prdm1 gene,

(2) Address the applicability of the Vitamin C enhancement of plasma cell differentiation by including data from additional culture conditions,

(3) Improve Discussion section of the manuscript.

*Reviewer #1 (Recommendations for the authors):*

Overall this is an interesting paper that describes a role for VC in augmenting B cell differentiation to PC. The authors have impressive data showing the effects in an in vitro system that works well in the mice and less well in human systems where there is more variation and small fold changes.

1) Deletion of the TET enzymes shows that the effects of VC are dependent in part on these enzymes. What other pathways are involved for the rest of the increase in PC formation?

2) As mentioned above, showing by ChIP that STAT3 binding is enhanced would close the circle of the premise. Without it the paper has a hole that needs filling.

3) I am surprised that the treatment with VC changes results in only 5 differentially expressed transcripts and no protein encoding genes, yet the DNA methylation pattern is disrupted during that time. Other studies have implied that there are changes in DNA methylation during B cell differentiation that are coupled to some extent with transcriptional changes. This should at least be discussed in more depth.

4) I am concerned as mentioned above about the interpretation of the changes in 5hmC following VC treatment. Naïve cells have the most 5hmC by dot dot blot and the heat map in all of the experiments. Most of the naïve sites are lost by Day 4 under either condition (SupFigure 4). To me it looks like the TET enzymes are inhibited and fail to continue to convert 5hmC to the downstream nucleosides either actively by TET or passive through replication (Figure 7A, D). A smaller number of sites accumulate 5hmC in the VC samples (SupFig4B), presumably by demethylating a 5mC->5hmC. Do the cells in your culture divide? If so, how could 5hmC be maintained? Is E58 in that upper grouping of new 5hmC sites (SupFig4B)? Please provide a clearer explanation of the interpretation/logic.

*Reviewer #2 (Recommendations for the authors):*

This paper is of interest to cell biologists and immunologists who study the molecular mechanisms underpinning the maturation of the antibody response. The work provides new insight into the overall activity of Vit C on B lymphocytes by showing that in these cells Vit C promotes differentiation to plasma cells by activation of Tet2/Te3 epigenetic elements with no contribution of its antioxidant activity. Additional experiments using Vit C oxidant inhibitors would strengthen the overall trust of the paper on the solely epigenetic impact of Vit C on B cell plasma cell differentiation through Tet2/Tet3 activation.

*Reviewer #3 (Recommendations for the authors):*

This paper titled "Epigenetic remodeling by Vit C potentiates the differentiation of mouse and human plasma cells" is a very interesting paper with novel findings and mechanisms of Vit C in plasma cell differentiation. They use a 2-step IL-4, CD40L, then IL-21 culture system of total B cells with and without Vit C to generate plasma cell secreting IgM, IgG1, and IgE in mice. They elegantly show that Vit C enhances plasma cell differentiation and that the mechanisms involve TET activity and DNA demethylation. These data may help to support Linus Pauling's claims that Vit C prevents and alleviates the common cold if the model can be applied broadly. Thus, the paper is interesting and a novel study. Overall, it is important and exciting.

Some questions should be addressed.

1. The mouse model of proliferation and differentiation is for IgM, IgG1 (Mouse) and IgE thus, is this also important for Type 1 immune responses such as viral infection? Is this a model of IgE plasma cell generation and thus Vit C exacerbates allergic disease or is this a model for all plasma cells IgE and not for IgG2a or IgG2b in mice and (IgG1 in human)?

2. Most of the cultures used total B cells from mice (which would have included memory and naive B cells). Interestingly, the cultures were taken for only 7 days. Later timepoints may have also been informative. Were later time points taken to provide ample time for naive B cell to plasma cell differentiation?

3. From the mouse data, there appeared to be a lot of variability in the results with Vit C. Some experiments showed 50% CD138+ plasma cell responses while others were as high as 80% CD138+ fractions. Could the authors explain?

4. The human experiments uses only naive B cells cultured with IL-21, CpG and antihuman CD40 or IL21 and pansorbin. Very few CD38 high cells were identified. Do they also have CD27 staining since CD19loCD38hiCD27hi have been important markers of early plasma cells? Is there more information about the isotypes by in the supernatants or elispots? Were there later time points for the plasma cell cultures to provide enough time for naive B cell to plasma cell differentiation?

5. The human responses appeared more modest and quite variable (with some subject's naive B cells with anti-CD40, CpG, and IL-21 did not increase at all). Pansorbin and IL-21 showed more consistent increases which were more modest. BCR stimulation is quite important for naive B cell activation. Is there a reason to use the stimulation described above? Do the authors have an explanation for this?

6. The assessment at the different phases of the cultures were interesting and important.

7. The TET studies are interesting and identifying the BLIMP-1 locus together with an enhancer site is interesting. With his expertise in the conditional CD19 cre TET mice, these experiments were clever and quite revealing. Are there other possible sites such as IRF4 or Xbp1 notable in these mice?

8. The paper reads well in the introduction and results. However, the discussion could be better written. The summary is helpful but additional thoughts of the implications of this study, limitations of interpretation due to the current in vitro stimulation models, and considerations of Vit C in memory versus naive B cell subsets, would have provided some context of this work.

---

## [Author Response]

Essential revisions:For revision, we would like the authors to focus on the following:(1) Provide ChIP-seq data on Stat3 binding in the Prdm1 gene,

We have performed STAT3 ChIP-seq using B cells stimulated with IL-21. The result showed that the STAT3 association with *Prdm1* 3’ enhancer E27 and its binding is increased in the presence of VC.

(2) Address the applicability of the Vitamin C enhancement of plasma cell differentiation by including data from additional culture conditions,

We have included additional culture conditions for both mouse and human B cells. For instance, we have shown that:

1. VC stereoisomer erythobic acid could also facilitate plasma cell differentiation (Figure 9—figure supplement 1-2);

2. Enzymatic oxidation of VC abolished the plasma cell enhancing activity (Figure 1—figure supplement 3);

3. VC could enhance plasma cell differentiation and antibody secretion in B cells stimulated with LPS and cytokines (Figure 1—figure supplement 4);

4. VC had a similar positive effect on plasma cell generation from polyclonal and monoclonal (BCR-knockin) naïve B cells (Figure 2—figure supplement 1);

5. In an extended 3-step culture, VC further increased the percentage of plasma cells that bear characteristics of mature plasma cells (Figure 2—figure supplement 3);

6. VC positively affects the differentiation of antibody secreting cells (or plasmablasts) from naïve human B cells that were stimulated with an addition condition (Figure 1—figure supplement 5).

(3) Improve Discussion section of the manuscript.

We have included additional discussions regarding the implication of our findings and the limitations of our study.

Reviewer #1 (Recommendations for the authors):Overall this is an interesting paper that describes a role for VC in augmenting B cell differentiation to PC. The authors have impressive data showing the effects in an in vitro system that works well in the mice and less well in human systems where there is more variation and small fold changes.1) Deletion of the TET enzymes shows that the effects of VC are dependent in part on these enzymes. What other pathways are involved for the rest of the increase in PC formation?

VC is an antioxidant and a co-factor for other 2-oxoglutarate-dependent dioxygenases (2OGDDs), including most histone demethylases. For instance, previous reports have demonstrated that VC can promote JHDM1a/1b (2OGDDs) to demethylate H3K36me2/3 and enhance the reprogramming of mouse embryonic fibroblasts to induced pluripotent stem cells (PMID 22100412). Another study showed that VC enhances KDM3a/b-dependent H3K9me2 demethylation in mouse embryonic stem cells (PMID 28706564). While our data showed that TET enzymes are necessary, it remains possible that other histone demethylases may be required for the effect of VC on PC formation.

Besides histone methylation, VC is implicated in the hypoxic response by enhancing the activity of prolyl hydroxylase domain (PHD) proteins that are also 2OGDDs (reviewed by PMID 25540771). In the presence of oxygen, PHD proteins hydroxylate the prolines on hypoxia-inducible factor-1/2 (HIF-1/2) and the proteasomemediated degradation of HIFs. Presumably, in the absence of VC, an increased HIF1/2 level may result in a hypoxic response despite the presence of oxygen. One study showed that hypoxia favors the in vitro differentiation of plasma cells from humans (28463531) and mice (27798169), suggesting that the absence of VC (increased HIF-1/2 levels) may promote PB/PC formation. How hypoxia may influence the positive effect of VC on PC in our culture system remains to be determined.

2) As mentioned above, showing by ChIP that STAT3 binding is enhanced would close the circle of the premise. Without it the paper has a hole that needs filling.

We really appreciate this comment that has led to our new discovery of another ascorbate-responsive element E27*.* While the previously proposed element E58 may regulate *Prdm1* expression, it is clear from our new ChIPseq data that STAT3 does not associate with E58 in B cells cultured in our system. Instead, STAT3 binds to E27 and the binding was enhanced by VC. Additional response to the comment was started above and is shown below again.

“We were unable to completely reproduce the previous T cell STAT3 ChIP-seq by Kwon *et al.*, 2009 (Dr. Warren Leonard’s lab) due to the discontinuation of the antibody (Santa Cruz). Therefore, we have tested four antiSTAT3 antibodies (Cell Signaling Technology rabbit monoclonal #4904, #12640, #9145; Millipore rabbit polyclonal #06-596), and found that only the rabbit mAb #9145 anti-phospho-Stat3 (Tyr705) from CST generated specific signals for ChIP, which was quantified by qPCR using an IL-21-induced positive control *Mcl1*.

Surprisingly, the ChIP-seq result showed that the STAT3 binding patterns differ between B cells and the previously published data from T cells (New Figure 8D and S8-1B). The difference may likely be due to cell types, culture conditions, and the source of antibodies. Nonetheless, our STAT3 ChIP-seq data showed that VC could enhance STAT3 binding at the *Prdm1* promoter and E27*,* a previously identified enhancer with a functional STAT3 motif at +27kb (New Figure 8D)(Kwon *et al.*, 2009). In order to prove that the DNA modification of this element was indeed regulated by VC, we analyzed the DNA methylation using bisulfite nanopore sequencing and showed that VC induced DNA demethylation at E27 (NEW Figure 9A). Importantly, both VC-enhanced DNA methylation (NEW Figure 9B) and STAT3 binding (NEW Figure 9C) at E27 were abolished in the *Tet2/3*deficient B cells. Our data strongly corroborate with the previously identified STAT3 binding site at E27 by Kwon *et al.*, where they showed that the STAT3 motif is responsible for IL-21-induced *Prdm1* expression. Our results further demonstrated that E27 is regulated by DNA methylation and is thus sensitive to the status of VC.”

3) I am surprised that the treatment with VC changes results in only 5 differentially expressed transcripts and no protein encoding genes, yet the DNA methylation pattern is disrupted during that time. Other studies have implied that there are changes in DNA methylation during B cell differentiation that are coupled to some extent with transcriptional changes. This should at least be discussed in more depth.

We agree with the Reviewer that the similar transcriptomes between B cells cultured with or without VC (Mock and VC) is surprising.

As mentioned by the Reviewer, most previous studies of plasma cell differentiation induced stimulation at the beginning of experiments (e.g., Shi et al., PMID 25894659) and observed substantial changes in the transcriptome. In another study (Barwick et al., PMID 27500631), the transcriptome and methylome also differ between naïve B cells and in vitro PB and PC after LPS stimulation. In fact, our results are consistent with the previous reports: we detected substantial changes in RNA expression after B cell activation (Figure 5A-B), accompanied by changes in the genome-wide distribution of 5hmC (Figure 7C-D).

As a DNA methylation intermediate, 5hmC often precedes the changes in DNA hypomethylation. The in vivo plasma cell differentiation experiments from Barwick et al., showed that the major changes in RNA expression and DNA methylation occur later in B cells that have undergone at least 8 cell divisions. Therefore, it is possible that the changes in 5hmC precede RNA expression and DNA methylation in our system.

We have included the following paragraph to the Discussion:

Line 399

The minimal difference in transcriptomes between control and VC groups on day 4 (Figure 5C) was surprising in light of the substantial effect of VC on plasma cell differentiation (Figure 1). The lack of difference in the transcriptome was likely not due to the sensitivity, as we were able to detect a significant difference in RNA expression between activated and naïve B cells (Figure 5A-B). A previous study reported the coupled changes in DNA methylation and RNA expression at the later stage of plasma cell differentiation^69^. Given 5hmC modification precedes DNA demethylation, we would expect 5hmC modification appear prior to transcriptional changes. Indeed, our results and another study on plasma cell differentiation^70^ support the notion that epigenome remodeling (e.g., 5hmC and chromatin accessibility) appears to precede transcriptional changes; and the changes in the epigenome may not necessarily result in immediate transcriptional responses.

4) I am concerned as mentioned above about the interpretation of the changes in 5hmC following VC treatment. Naïve cells have the most 5hmC by dot dot blot and the heat map in all of the experiments. Most of the naïve sites are lost by Day 4 under either condition (SupFigure 4). To me it looks like the TET enzymes are inhibited and fail to continue to convert 5hmC to the downstream nucleosides either actively by TET or passive through replication (Figure 7A, D). A smaller number of sites accumulate 5hmC in the VC samples (SupFig4B), presumably by demethylating a 5mC->5hmC. Do the cells in your culture divide? If so, how could 5hmC be maintained? Is E58 in that upper grouping of new 5hmC sites (SupFig4B)? Please provide a clearer explanation of the interpretation/logic.

As the Reviewer suggested above, we have included a better explanation of how TET functions and the fate of 5hmC after DNA replication (Figure 7—figure supplement 1). During DNA replication, the 5mC at CpG motif will pair with the unmodified cytosine on the newly synthesized complementary strand, creating a “hemi-methylated CpG”. Hemi-methylated CpG is recognized by the maintenance DNA methyltransferase complex DNMT1 and UHRF1, which is required for methylating the unmodified cytosine at the hemi-methylated CpG. However, the 5hmC-containing CpG motif is not recognized by the DNMT1 complex, and thus the CpG on the newly synthesized strand is not methylated. As a result, after rounds of cell division, the percentage of 5hmC on the original DNA will be diluted by the newly synthesized DNA. This explains why naïve B cells have more 5hmC than the activated B cells and lose most of the 5hmC peaks after divisions. In fact, in the 40LB co-culture, naïve B cells expand ~40 times after four days of culture, equating to around five divisions.

Since 5hmC is not replicated, all newly gained 5hmC are generated by TET-mediated oxidation of 5mC. Therefore, in Figure 7—figure supplement 2, those VC^up^ regions (Figure 7-figure supplement 2, top panel) represent de novo 5mC oxidation into 5hmC by TET enzymes that are presumably recruited by TFs. Meanwhile, the common regions (Figure 7—figure supplement 2, middle panel) are likely caused by the continuous recruitment of TET at the same regions that are enriched in 5hmC in naïve B cells. A recent study has demonstrated that the de novo DNA methyltransferase DNMT3A/B and TET can be simultaneously recruited to certain loci, where the cytosines undergo constant methylation turnover (32471981). We speculate that these common regions may represent these cytosines that recruit both DNMT3A/B and TET. Therefore, after the CpG on newly synthesized DNA is not methylated by DNMT1, the CpG is later methylated by DNMT3A/B, which are capable of methylating unmodified cytosines, and then oxidized by TET. It is still currently unknown how 5hmC is the preferential oxidative product of TET, which can oxidize 5hmC further into other minor oxidized cytosines (31965999).

Reviewer #2 (Recommendations for the authors):This paper is of interest to cell biologists and immunologists who study the molecular mechanisms underpinning the maturation of the antibody response. The work provides new insight into the overall activity of Vit C on B lymphocytes by showing that in these cells Vit C promotes differentiation to plasma cells by activation of Tet2/Te3 epigenetic elements with no contribution of its antioxidant activity. Additional experiments using Vit C oxidant inhibitors would strengthen the overall trust of the paper on the solely epigenetic impact of Vit C on B cell plasma cell differentiation through Tet2/Tet3 activation.

We would like to thank you for the suggestion. We have performed additional experiments to explore the mechanism of how VC may affect plasma cell (PC) differentiation using our system. Note that in most experiments, the redox-stable version of VC 2-phospho-ascorbic acid (P-AA) was used. In the experiments below, both P-AA and L-ascorbic acid (L-AA) were used and they were collectively represented by VC.

First, we have explored whether erythobic acid (EA), a stereoisomer of L-AA, may have the same activity on PC (NEW Figure 1—figure supplement 2). EA differs from L-AA by the orientation of the hydroxyl group on the 5^th^ carbon. Our results showed that EA can promote PC differentiation with a lower activity compared to L-AA. This data suggest that the conformation of VC is not absolutely required for the PC enhancing activity.

Second, we tested whether oxidized L-AA is able to promote PC differentiation. To generate dehydroascorbic acid (DHA) from L-AA, we oxidized L-AA with ascorbate oxidase (AAO) in vitro prior to adding the reactions to B cell cultures. We found that AAO oxidation diminished the effect of L-AA on PC differentiation (NEW Figure 1figure supplement 3). As a control, the AAO-resistant P-AA was not affected by AAO (NEW Figure 1—figure supplement 3). These results demonstrated that the oxidation of VC inhibits its ability to facilitate the differentiation of PC.

Reviewer #3 (Recommendations for the authors):This paper titled "Epigenetic remodeling by Vit C potentiates the differentiation of mouse and human plasma cells" is a very interesting paper with novel findings and mechanisms of Vit C in plasma cell differentiation. They use a 2-step IL-4, CD40L, then IL-21 culture system of total B cells with and without Vit C to generate plasma cell secreting IgM, IgG1, and IgE in mice. They elegantly show that Vit C enhances plasma cell differentiation and that the mechanisms involve TET activity and DNA demethylation. These data may help to support Linus Pauling's claims that Vit C prevents and alleviates the common cold if the model can be applied broadly. Thus, the paper is interesting and a novel study. Overall, it is important and exciting.

Thank you very much for the comments and suggestions. We have performed additional experiments and have strengthened the discussion. Please see below for the point-to-point response.

Some questions should be addressed.1. The mouse model of proliferation and differentiation is for IgM, IgG1 (Mouse) and IgE thus, is this also important for Type 1 immune responses such as viral infection?

As noted by the Reviewer, the 40LB co-culture system generated Type 2 associated isotypes (IgG1, IgE). We have attempted to induce other isotypes using the 40LB system (with TGF-b or with IFN-g) but were not successful. Therefore, to induce the switching to other isotypes, we stimulated mouse B cells with LPS and a combination of cytokines following a previous study (PMID 11007474), including LPS alone (for IgM and IgG3), LPS+IFN-g (IgG2c), and LPS+IL-4 (IgG1). Similar to the observation using 40LB co-culture, the addition of VC increased the percentage of plasma cells and the secretion of antibodies with multiple isotypes (NEW Figure 1figure supplement 4). The lower VC-dependency in the LPS+/-cytokine culture compared to 40LB could potentially be attributed to multiple factors, including different signaling pathways (TLR vs CD40/BAFFR) and the lower proliferation rate (thus lower passive DNA demethylation). Nonetheless, our results showed that VC is able to promote plasma cell differentiation regardless of the switched isotypes.

Is this a model of IgE plasma cell generation and thus Vit C exacerbates allergic disease or is this a model for all plasma cells IgE and not for IgG2a or IgG2b in mice and (IgG1 in human)?

Unfortunately, we have not examined the role of VC using an allergy model. However, we speculate that in vivo transfer of IgE-secreting antigen-specific plasma cells generated with the 40LB system will likely exacerbate allergy or asthma pathology.

2. Most of the cultures used total B cells from mice (which would have included memory and naive B cells). Interestingly, the cultures were taken for only 7 days. Later timepoints may have also been informative. Were later time points taken to provide ample time for naive B cell to plasma cell differentiation?

We appreciate the insightful comments.

While the mice we used are between 6-12 weeks old, it is possible to have memory B cells in the starting population. To mitigate this concern about memory B cells, we isolated B cells from IgHCGG mice, a strain expressing BCR-specific for chicken γ globulin (CGG) generated by Dr. Gabriel Victora (PMID 30181412). In IgHCGG, the IgH and Igk from a CGG-specific antibody were joint and knocked into the IgH locus. Around 80% of CD19^+^ B cells express the CGG-specific receptor, the expression of which restricts the recombination of endogenous IgH and Igk/l. Therefore, in the absence of cognate antigen CGG, most follicular B cells express IgM and presumably to have a naïve phenotype. We isolated the B cells from B6 and IgHCGG and cultured them using the 40LB system as in Figure 1. The result showed no significant difference between the two groups (NEW Figure 2—figure supplement 1). This result suggests that VC can enhance plasma cell generation from naïve mouse B cells. Whether VC has a differential effect on memory cells remains to be determined.

For the human B cells experiment, we also provided data to verify the purity of naïve B cells (NEW Figure 1figure supplement 5). The typical purity is >95%. Whether human memory B cells may have different requirements for VC remained to be studied but were discussed (in Discussion).

To address the comment about the later timepoint, we have cultured the B cells for an extended period (11 days) with a 3-step culture (NEW Figure 2—figure supplement 3A). Consistent with the day 7 results (Figure 1), a significantly higher percentage of plasma cells were generated in the presence of VC (NEW Figure 2—figure supplement 3B), suggesting VC may not simply accelerate the kinetics of plasma cell generation. Interestingly, the majority of the CD138^+^ plasma cells from the VC group downregulated CD19 (NEW Figure 2—figure supplement 3C), suggesting these CD138^+^ cells may have a higher resemblance to in vivo mature plasma cells.

3. From the mouse data, there appeared to be a lot of variability in the results with Vit C. Some experiments showed 50% CD138+ plasma cell responses while others were as high as 80% CD138+ fractions. Could the authors explain?

Thank you for pointing out the inherent variability. While the intraexperiment variability is very low, the variability is occasionally higher between experiments. We attempted to identify the source of the variability and found that the condition of 40LB (>10 passages) may affect the outcome. It is possible that the 40LB cells decrease the expression of CD40L and BAFF after extended culture and thus resulting in lower overall plasma cell generation.

Another source of variability potentially comes from the small variation in the B cell seeding density. We have tested and found that higher starting cell numbers during 1^st^ culture usually resulted in an overall lower plasma cell generation after 2^nd^ culture. One possibility is the competition between B cells for CD40L and BAFF. Another possibility is that B cells may produce factor(s) that decrease the differentiation of plasma cells.

4. The human experiments uses only naive B cells cultured with IL-21, CpG and antihuman CD40 or IL21 and pansorbin. Very few CD38 high cells were identified. Do they also have CD27 staining since CD19loCD38hiCD27hi have been important markers of early plasma cells? Is there more information about the isotypes by in the supernatants or elispots? Were there later time points for the plasma cell cultures to provide enough time for naive B cell to plasma cell differentiation?

Unfortunately, CD27 was not included in the initial staining panel as we followed the protocol and gating scheme from an earlier publication that included IL-21 in the plasma cell culture (PMID 16339522). However, we have performed an additional stimulation condition and have incorporated CD27 in the analysis (NEW Figure 2—figure supplement 5B-C). Consistent with the earlier results (Figure 1F-G), VC could enhance plasma cell differentiation in the 3^rd^ culture condition. A similar result was observed when using CD38^hi^CD27^+^ as the marker for early plasma cells. These early plasma cells cultured with VC had a lower surface expression of CD19. Longer team culture may be needed to observe the complete CD19 downregulation and the expression of additional plasma cell features.

We have not performed ELISA or ELISPOT with the human samples. As noted above, we have performed a later time point culture with mouse cells (NEW Figure 2—figure supplement 3) but have not been successful in longer-term human B cell culture.

5. The human responses appeared more modest and quite variable (with some subject's naive B cells with anti-CD40, CpG, and IL-21 did not increase at all). Pansorbin and IL-21 showed more consistent increases which were more modest. BCR stimulation is quite important for naive B cell activation. Is there a reason to use the stimulation described above? Do the authors have an explanation for this?

The main reasons for using these conditions were based on previous publications (PMID 16339522) where they have used IL-21 in their culture system. Initially, we tested multiple conditions from the study and attempted to include conditions that involved both CD40 or BCR stimulations. However, we were not able to reliably generate antigen secreting cells using anti-IgM-F(ab')2 due to abnormally high cell death. Since pansorbin (*S. aureus* Cowan) has a high level of Protein A on the bacteria surface, we reasoned that it would trigger BCR signals via Protein A-Ig crosslinking (PMID 6974188) in addition to activating multiple TLRs. Therefore, we chose the two conditions shown in Figure 1 where CD40+CpG+IL-21 mimics the T-dependent stimulation, while pansorbin+IL21 (triggering BCR and TLRs) mimics T-independent stimulation.

We have investigated the reason behind the lack of increase after VC culture for some subjects. See Author response image 1 for the breakdown of the results by experiment. Each experiment was performed with PBMC from non-overlapping donors. We found that the responsiveness to VC varied among individual donors. Experiments 3 and 4 have low overall ASC differentiation and responsiveness to VC (Author response image 1). Further investigations are needed to identify the factors that have contributed to the variability.

**Author response image 1. sa2fig1:** The responses of human B cells to VC vary among individuals. (A) Stimulated with CpG (ODN) anti-CD40 and IL-21; (B) Stimulated with SAC and IL-21. (Corresponding to Figure 1F and 1G).

6. The assessment at the different phases of the cultures were interesting and important.

Thank you very much for the compliment.

7. The TET studies are interesting and identifying the BLIMP-1 locus together with an enhancer site is interesting. With his expertise in the conditional CD19 cre TET mice, these experiments were clever and quite revealing. Are there other possible sites such as IRF4 or Xbp1 notable in these mice?

We really appreciate the positive comments. While the expression of *Irf4* and *Xbp1* did not change significantly in B cells activated for four days in the presence of VC, we did detect significant 5hmC modification changes at the loci (Author response image 2). For instance, at *Xbp1* locus (left), VC induced an additional intronic 5hmC modification. The common 5hmC peaks between naïve and Mock^Day4^ likely undergo demethylation in VCDay4, a scenario similar to E27 at *Prdm1.* Similarly, VC is required for 5hmC modification at *Irf4* locus (right). Most importantly, VC similarly facilitates STAT3 association at *Xbp1* promoter and *Irf4* gene. The role of VC on genome-wide STAT3 binding will be the subject of further study.

**Author response image 2. sa2fig2:** VC mediated 5hmC-modification at *Xbp1* and *Irf4* loci. Genome browser tracks of the 5hmC enrichment at *Xbp1* and *Irf4* loci (mm10). Top panels: VC enhanced STAT3 association at *Xbp1 and Irf4*. B cells were cultured with or without VC for 4 days as in Figure 4 and treated with or without rmIL-21 (10 ng/mL) for 6h. STAT3 binding was analyzed by ChIPseq. The signals on the genome browser tracks are the average from two biological replicates. Data shown are the average of two biological replicates. Middle panels: 5hmC distribution in naïve B cells (Naïve), day 4 activated control B cells (Mock^Day4^), and VCtreated B cells (VC^Day4^). Data shown are the average of two biological replicates.Bottom panel: Differential 5hmC-enriched regions (DhmRs) between the indicated group as in Figure 7C. Colors indicate the differential status of the regions as depicted in the legend.

8. The paper reads well in the introduction and results. However, the discussion could be better written. The summary is helpful but additional thoughts of the implications of this study, limitations of interpretation due to the current in vitro stimulation models, and considerations of Vit C in memory versus naive B cell subsets, would have provided some context of this work.

Thank you for the constructive comments. We have added included additional discussion (listed below).

“DNA methylation is a dynamic process and undergoes substantial remodeling during differentiation. Studies have shown that the differentiation from naïve B cells to plasma cells and memory B cells is accompanied by the remodeling of methylome, with more regions being hypomethylated that is consistent with the role of TET enzymes in cell differentiations^74-76^. Therefore, these TET-dependent differentiations may be influenced by the levels of VC. We speculated that methylome remodeling may partially contribute to the increased effectiveness of memory B cells to differentiate into germinal center B cells and plasma cells. Interestingly, a previous report showed that human memory B cells, compared to naïve B cells, require a lower level of STAT3 to differentiate into plasma cells^77^. As DNA hypomethylation may enhance STAT3 binding, DNA demethylation may contribute to the lower requirement for STAT3 in memory B cells. In addition, given the lower methylation levels in memory compared to naïve B cells, we hypothesize that the differentiation of plasma cells from naïve B cells may be preferentially affected by VC deficiency.

VC has long been associated with immune function but the notion continues to be controversial. While VC supplementation does not prevent “common cold”, a comprehensive meta-analysis of 29 clinical trials showed that VC supplementation is associated with shortened duration of symptoms^78^. However, how VC may enhance anti-viral immunity remained unclear. Giving the importance of early IgM produced by antibody secreting cells in controlling infection^79^, it is possible that VC deficiency may delay the plasma cell differentiation to result in inferior control of the viruses at the early phase. Here, our in vitro findings provided a mechanistic explanation of how VC may affect humoral immune response. It is currently unclear how VC-deficiency affects B cell function in vivo. Future experiments using genetic VC-deficient animal models (hamsters or *Gulo-*deficient mice) will provide critical insight into the physiological function of VC in immune response.”